# Multiphoton excited singlet/triplet mixed self-trapped exciton emission

Rui Zhou[1,7], Laizhi Sui[2,7], Xinbao Liu[3], Kaikai Liu [1]✉, Dengyang Guo[1,4], Wenbo Zhao[1], Shiyu Song[1], Chaofan Lv[1], Shu Chen[1], Tianci Jiang[5,6], Zhe Cheng[5,6], Sheng Meng [3] & Chongxin Shan [1]✉

Multiphoton excited luminescence is of paramount importance in the field of optical detection and biological photonics. Self-trapped exciton (STE) emission with self-absorption-free advantages provide a choice for multiphoton excited luminescence. Herein, multiphoton excited singlet/triplet mixed STE emission with a large full width at half-maximum (617 meV) and Stokes shift (1.29 eV) has been demonstrated in single-crystalline ZnO nanocrystals. Temperature dependent steady state, transient state and time-resolved electron spin resonance spectra demonstrate a mixture of singlet (63%) and triplet (37%) mixed STE emission, which contributes to a high photoluminescence quantum yield (60.5%). First-principles calculations suggest 48.34 meV energy per exciton stored by phonons in the distorted lattice of excited states, and 58 meV singlet-triplet splitting energy for the nanocrystals being consistent with the experimental measurements. The model clarifies long and controversial debates on ZnO emission in visible region, and the multiphoton excited singlet/triplet mixed STE emission is also observed.

Multiphoton excited luminescence is a ubiquitous phenomenon that an emissive material absorbs multiple photons of low energy simultaneously and emits a photon of high energy under high-intensity incident laser light[1]. The initial multiphoton process was theoretically predicted in 1931, and the two-photon excited up-conversion emission was confirmed experimentally in 1961 in $CaF_2$:$Eu^{2+}$ crystals under a pulsed ruby laser excitation[2]. Since then, three-photon[3], four-photon[4] or even higher-order multiphoton emission related processes[5,6] have been observed with the emergency of femtosecond laser. Multiphoton related studies include both fundamental research (such as multiphoton-induced surface photoelectric effect[1], multiphoton-induced photochemical reactions[1] and multiphoton spectroscopy[7]) and applications of multiphoton absorption, multiphoton excitation, and multiphoton active materials (including microfabrication[8], three-

dimensional data storage[8] and imaging[9]). These studies may greatly enrich and deepen our knowledge and understanding of the interactions between strong coherent radiation and matter, and have attracted a wide attention in literature[1]. Multiphoton imaging technology possessing higher spatiotemporal resolutions, deeper penetration, diminished tissue autofluorescence interference, and reduced phototoxicity or photodamage to biological samples has been extensively researched[10]. Self-trapped exciton (STE) emission with outstanding optical property and self-absorption free advantages provides an excellent choice for multiphoton excited luminescence.

STE emission has been widely studied in recent years for its intriguing photonics properties and potential application[11–16]. STEs usually occur in strong electron-phonon coupling systems, in which elastic distortions of the lattice will be caused by the surrounding

[1]Henan Key Laboratory of Diamond Optoelectronic Materials and Devices, School of Physics and Microelectronics, Zhengzhou University, Zhengzhou, P. R. China. [2]State Key Laboratory of Molecular Reaction Dynamics and Dalian Coherent Light Source, Dalian Institute of Chemical Physics, Chinese Academy of Sciences, Dalian, P. R. China. [3]Institute of Physics, Chinese Academy of Sciences, Beijing, China. [4]Department of Physics, Cavendish Laboratory, University of Cambridge, Cambridgeshire, UK. [5]Department of Pulmonary and Critical Care Medicine, The First Affiliated Hospital of Zhengzhou University, Zhengzhou, P. R. China. [6]Henan Key Laboratory for Pharmacology of Liver Diseases, Zhengzhou, P. R. China. [7]These authors contributed equally: Rui Zhou, Laizhi Sui. ✉e-mail: liukaikai@zzu.edu.cn; cxshan@zzu.edu.cn

excited electrons and holes[17,18]. In this case, excited electrons and holes will be trapped by self-trapped states induced by the lattice distortions quickly, due to their lower energy compared with free carrier states[19]. Recently, several kinds of materials with STE emission have been reported, including halide perovskites[20], condensed rare gases[21] and organic materials[22]. For example, Luo et al. achieved significant improvement in PL QY of a $(F_2CHCH_2NH_3)_2CdBr_4$ perovskite from <1% to 32.5% by alloying these two isostructural perovskites and demonstrated the intriguing effect of alloying on activating STE emission as an effective approach to control and enhance the optical properties of metal halide perovskites[23]. Quan et al. reported STE emission from 0D $Cs_3Cu_2Cl_5$ nanocrystals with a high PL QY of 48.7%, broadband blue-green emission and large Stokes shifts[24]. Tang et al. reported that warm-white light emission with PL QY of 86 ± 5% could be achieved from $Cs_2(Ag_{0.60}Na_{0.40})$ $InCl_6$ with 0.04% bismuth doping[17]. The broad emission spectrum, large stokes shift, small self-absorption and high PL QY endow STE materials with unique advantages in the lighting and displaying. The mentioned merits make them also promising candidates in bioscience and optoelectronics. Excitons refer to coupled electrons and holes through Coulomb interaction, the energy depends on their spin configuration[25]. It is known that electrons and holes are fermions, two fermions cannot occupy the same state according to Pauli exclusion principle[26]. That is, the sign of wave function must change if interchanging coordinates. The spin of carriers may flip due to strong orbital motion and spin of carriers. Thus, the total angular momentum of one exciton can be 0 or 1, which corresponds to singlet or triplet excitons, and can be described with four wave functions[20,26]. Triplet excitons have lower energy compared with singlet excitons, but they are transition forbidden[20,27]. However, strong spin-orbit coupling

in conduction band can induce emission of triplet excitons, as demonstrated by Becker et al.[26]. Thus, realizing radiative recombination of singlet and triplet mixed excitons is an effective method to increase the emission efficiency. However, most of these reported materials with STE emission are perovskite-based materials, the toxic elements, unstable and water-insoluble properties limit their applications in the field of bio-photonics drastically. ZnO nanoparticles (NPs) as an eco-friendly nanomaterial with visible light emission have a very long and complex history, they are widely considered to be less than stellar due to the low PL QY and defect induced emission mechanism.

In this work multiphoton excited singlet/triplet mixed STE emission with the high PL QY of 60.5% has been achieved. First-principles theoretical calculations of excited state properties confirm that the emission of ZnO nanocrystals is originated from the radiative recombination of singlet/triplet mixed STE. The ZnO nanocrystals have good water dispersibility, emission stability and biocompatibility. In addition, multiphoton excited emission is observed under excitation of 800–1600 nm, and the multiphoton bioimaging have been demonstrated to record 3D information in mice muscle as deep as 850 μm. This work will spark new applications of STE emission materials in the field of multiphoton bioimaging.

## Results

### Structure and STE emission investigation

A ZnO nanocrystal synthesized through chemical method is shown in Fig. 1a, the ZnO nanocrystal show bright yellow emission under excitation of UV and NIR light, including single photon and multiphoton excited emission. The emission spectrum includes free exciton (FE) and STE emission with photon energy centered at around 3.32 eV and

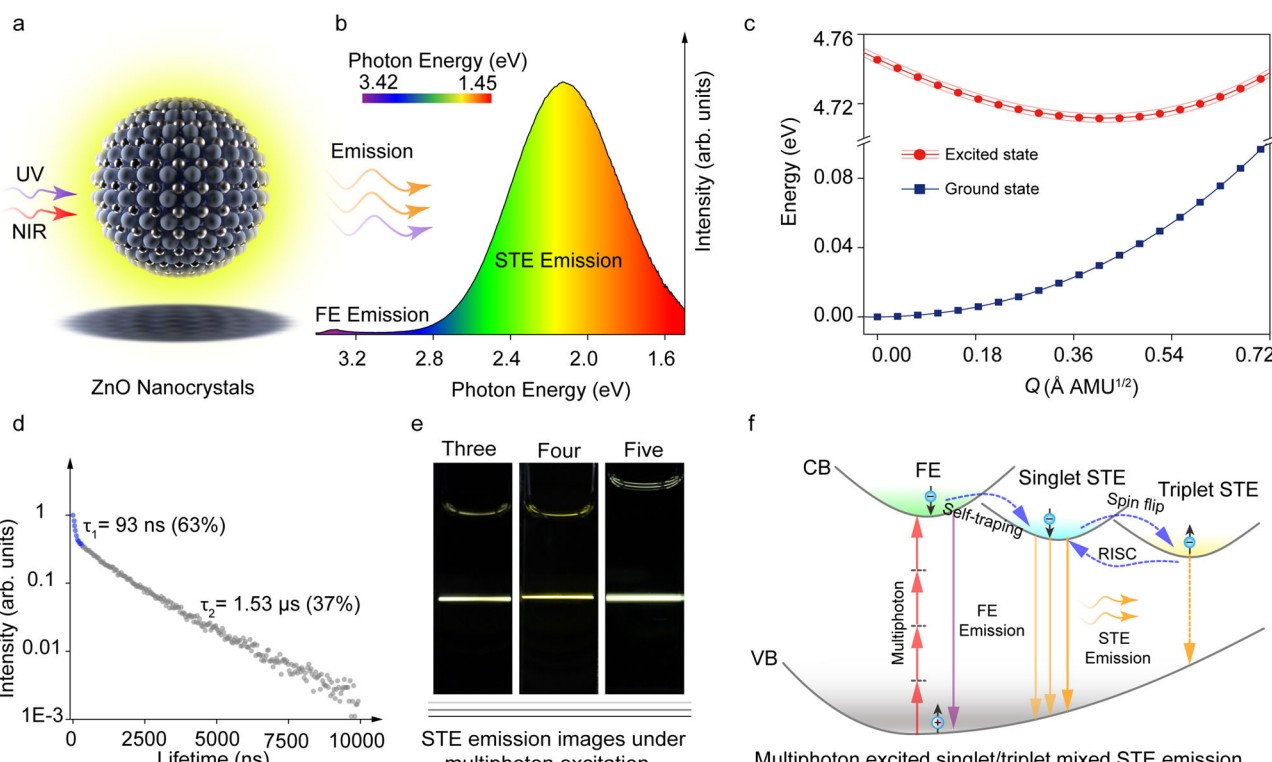

**Fig. 1 | STE emission in ZnO nanocrystals and a schematic representation of multiphoton excited singlet/triplet mixed STE emission. a** A schematic diagram of a ZnO nanocrystal under excitation of UV and NIR light. **b** Emission spectrum of the ZnO nanocrystals at room temperature. **c** Configuration coordinate diagram for the STE formation in ZnO nanocrystals. **d** PL decay curves of the ZnO nanocrystals under 375 nm excitation at room temperature, 63% excitons are annihilated within 93 ns and 37% excitons are annihilated within 1.53 μs. **e** Digital photos of

three-photon, four-photon, and five-photon excited emission of the ZnO nanocrystals under 800 nm, 1200 nm and 1600 nm fs laser at room temperature. **f** Schematic illustrating the multiphoton excited singlet/triplet mixed STE emission process in the ZnO nanocrystals. (NIR light near infrared light, Ex excitation, Em emission, CB conduction band, VB valence band, FE free exciton, ISC intersystem crossing, RISC reverse intersystem crossing).

2.1 eV (Fig. 1b). The FE emission can be ignored comparing with STE emission, indicating STE emission play a key role in the nanocrystals. The STE emission shows a broad spectrum with energy difference of about 1.4 eV. First principles density-functional theory calculation with PBE0 functional was used to investigate the origin of STE emission qualitatively based on previous benchmarks of the band gap and lattice constant[28–30], as shown in Fig. 1c. Our calculation indicated that the lattice distortion occurred after photoexcitation, causing the shift in atomic coordinates of the lowest energy of excited state, which was beneficial to the generation of broad spectrum. The PL decay curves of the ZnO nanocrystals show a bi-exponential form with 63% of the population decaying with a 93 ns lifetime, and the remainder with a 1.53 μs lifetime (Fig. 1d). The fast decay component is attributed to the direct decay from the singlet STE state, while the appearance of a slow decaying component is related to triplet STE state, this will be discussed carefully in the next part. The synthesized ZnO nanocrystals also show multiphoton absorption ability, and can emit photons under excitation of 800–1600 nm. The digital photos of three-photon, four-photon, and five-photon excited emission of ZnO nanocrystals under 800 nm, 1200 nm and 1600 nm fs laser in aqueous solution are showed in Fig. 1e. The multiphoton excited singlet/triplet mixed STE process is schematically shown in Fig. 1f. The electrons in ground states are excited into excited states, forming free carriers. Due to the strong electron-phonon coupling in the ZnO nanocrystals, light excitation can induce lattice distortion, which traps free carriers to form STEs[18]. Spin-orbit coupling within the nanocrystals can lead to electron spin flip, thus, STEs include singlet/triplet mixed excitons[16]. Singlet STEs emit photons through fast decay process, while the triplet STEs return to singlet through reverse intersystem crossing and then emit photons.

Broad spectrum emission in visible light range as typical feature is commonly observed in ZnO, while emission in ultraviolet ranges is the focus of researchers in the past decades[31]. The emission in visible region is usually attributed to the recombination of deep defect levels, low PL QY and possible defects in ZnO are the reasons of researchers' judgement[32,33]. However, the emission of ZnO in visible range is a topic rife with controversies until now. In this work, ZnO nanocrystals with high crystallinity were synthesized, and the average PL QY of the nanocrystals can reach as high as 60.5%. The ZnO nanocrystals show bright emission in solid or liquid state under excitation of UV light, and the corresponding images are shown in Fig. S1. The structure and morphology of the ZnO nanocrystals are characterized carefully, and the high-resolution scanning transmission electron microscopy (HR-STEM) images of ZnO nanocrystals viewed along [110] and [001] zone axis are shown in Fig. 2a, notably, the nanocrystals present high integrity in crystal structure. The interplanar spacing with 1.6 Å, 2.6 Å and 2.8 Å can be observed clearly, which corresponds to (110), (002) and (010) facets[34,35]. The corresponding selected-area diffraction images also indicate the high crystallinity of the ZnO nanocrystals, which agree well with the simulated results (right of Fig. 2a). Thus, although the defects from surface boundaries can influence the optical performance of the ZnO nanocrystal, the surface silica can passivate the surface defects greatly based on our previous work, FTIR and XPS spectrum (Fig. S2)[36]. In addition to transmission electron microscope (TEM), atomic force microscope (AFM) images, dynamic light scattering spectrum and X-ray diffraction (XRD) spectrum indicate the uniformity of the prepared ZnO nanocrystals (Fig. S3).

In order to investigate emission mechanism, optical properties of the ZnO nanocrystals were investigated from all aspects. The absorption, PL excitation and emission spectra are integrated into Fig. 2b. Large Stokes shift (225 nm, 1.29 eV) and broad spectrum (full width at half-maximum (FWHM) of 177 nm, 617 meV) are observed, which is regarded as typical feature of STE emission[18]. The absolute PL QY of the prepared ZnO nanocrystals before and after coating silica was measured for five times to ensure the accuracy, as shown in Fig. 2c. The

average absolute PL QYs of ZnO nanocrystals without and with silicon were 55.2% and 60.5%. The PL QYs of ZnO nanocrystals do not increase or decrease significantly after coating the silicon shell, indicating that the surface states do not play a dominant role in the broad PL emission of ZnO nanocrystals. 3D contour plots spectra of emission versus excitation are shown in Fig. 2d, broad emission range and excitation-independent characterizations are clearly observed. For emission from 400 to 850 nm, the excitation spectra have identical shapes and features, suggesting that the broad emission originates from the recombination of the same initial excited state[17]. Additionally, light with wavelength over 400 nm cannot induce emission of the ZnO nanocrystals in visible region from PL excitation spectrum, indicating that excitation energy below-exciton or sub-gap do not contribute to the emission from permanent defects[17]. Furthermore, it has been proved that excitonic emission depends linearly and slightly super-linearly on the excitation intensity, but the defects-induced emission exhibits sublinear behavior due to the limited number of the defect states[18]. The broad emission from ZnO nanocrystals versus excitation powder over 3 orders of magnitude was recorded (Fig. 2e), and the emission intensity shows linear dependence on the excitation intensity, also indicating that the broad emission does not arise from permanent defects[19]. The average ratio of Zn to O atoms in three ZnO nanocrystals samples without silica coating by XPS measurement was 0.99, which was close to 1 (Table S1). In addition, electron paramagnetic resonance (EPR) spectroscopy is one of an effective method for determining the presence of impurities and native defects in solid state materials[31]. The EPR signal with g-factor close to the free-electron value (2.0023) is generally due to an unpaired electron trapped on an oxygen vacancy site (1.9965, 1.9948, 2.0190 or 2.0106)[31]. In our case, no obvious EPR signal in the self-prepared ZnO nanocrystals is observed (Fig. S4). This indicate that the surface of ZnO nanocrystals should be oxygen-rich. In addition, temperature-dependent PL of the ZnO nanocrystals was conducted to determine the electron-phonon coupling and Huang-Rhys factor ($S$), as shown in Fig. 2f and 2g. The emission intensity increases when temperature decreases from 300 to 80 K, along with a decrease in the full width at half-maximum (FWHM). This is attributed to the suppression of non-radiative pathways at lower temperature. The calculated STE binding energy is ~ 98.27 meV (Fig. S5), which is close to lead-based and lead-free perovskites (~50–100 meV)[23] and conductive to the radiative recombination processes. The FWHM of PL spectra versus temperature is shown in Fig. 2g, and electron-phonon coupling was assessed based on the relationship between FWHM of PL spectra by the below Eq. (1)[17]:

$$FWHM(T) = 2.36\sqrt{S}\hbar\omega_{phonon}\sqrt{coth(\hbar\omega_{phonon}/2k_B T)} \qquad (1)$$

where $\omega_{phonon}$ is phonon frequency, $\hbar$ is reduced Planck constant, $T$ is temperature, and $k_B$ is Boltzmann constant. In general, Huang-Rhys factor $S$ can be calculated by fitting the temperature-dependent FWHM with Eq. 1. The degree of fitting (R-squared) is 0.99, indicating the fitted result is reliable. Huang-Rhys factor $S$ value of 30.8 is thus obtained, comparable to that of perovskites and large enough for STE emission (Table S2), and this is the reason of broad emission spectra[19]. With the phonon energy $\hbar\omega_{phonon}$ = 35.8 ± 1.24 meV, and the time for the free excitons to be trapped into STEs ($\tau = 2\pi/\omega_{phonon}$) can be calculated as 115.3 fs, which indicates an ultrafast transition from free excitons to STEs following photoexcitation[17]. Remarkably, the corresponding LO mode in our theoretical calculation ($\hbar\omega$ = 48.34 meV) shows a resemble pattern with the atomic distortion under photoexcitation. The eigenvectors of LO mode correspond to the direction of bond length stretch, as shown in Fig. 2h, indicating longitudinal optical photon plays a dominant role for line width broadening (as illustrated in the right of Fig. 2h). Our calculation also shows that the bond stretching results from the charge transfer from the $p$-orbital of O atoms to the $s$-orbital of Zn atoms (Fig. S6), which may be an antibonding state here.

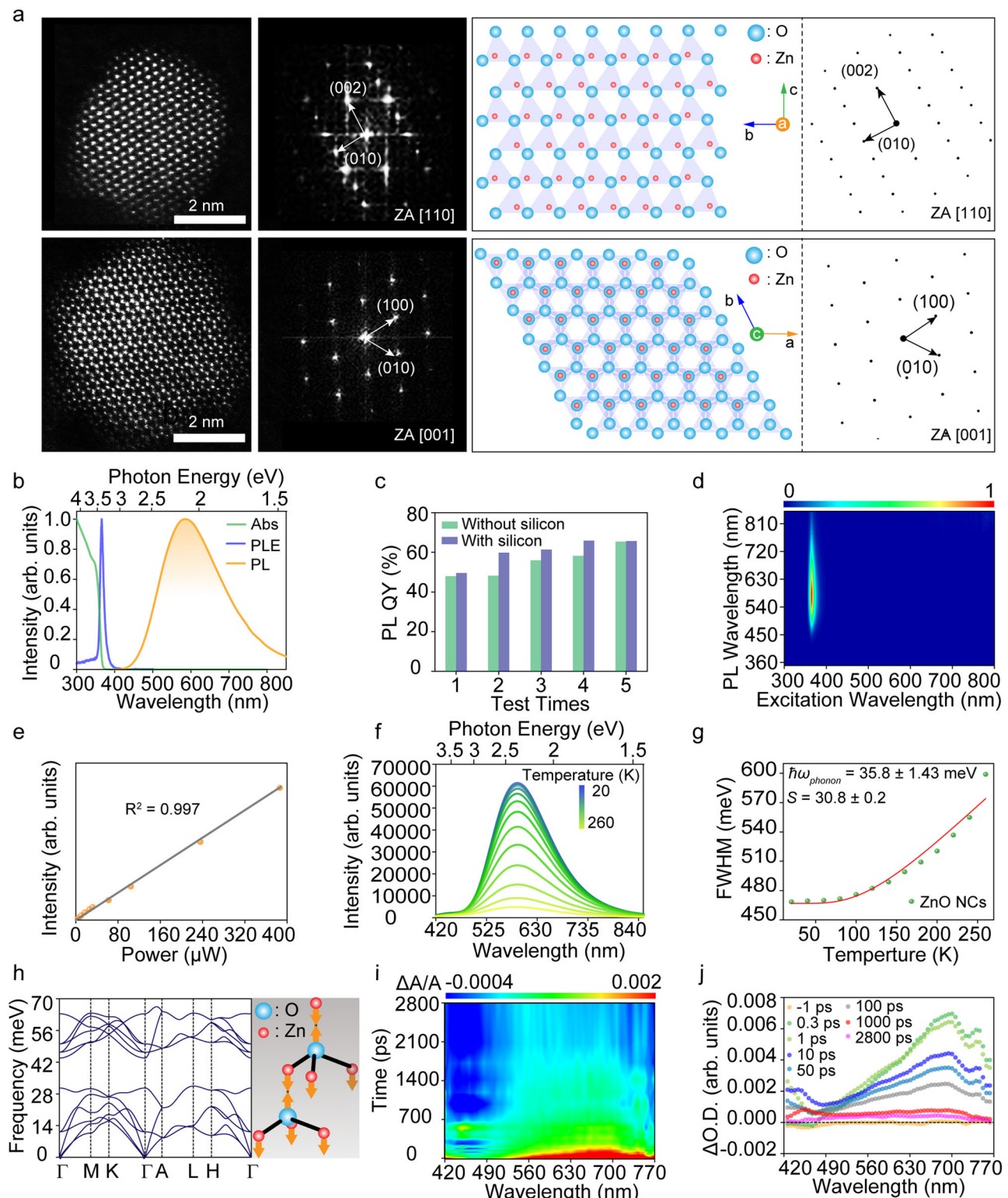

**Fig. 2 | Structure and optical properties of the ZnO nanocrystals. a** HR-STEM images of the ZnO nanocrystals viewed along [110] and [001] zone axis, and corresponding selected-area diffraction images. **b** Absorption, PL and PL excitation spectra for the ZnO nanocrystals at room temperature. **c** PL QY values of the ZnO nanocrystals without and with silica coating tested for five times at room temperature. **d** Emission contour plots of the ZnO nanocrystals as excitation wavelength varies from 250–800 nm at room temperature. **e** Emission intensity versus excitation power for the ZnO nanocrystals at room temperature (excitation power:

0.3–386.125 μW, yellow symbols), linear fit result has a high R² value of 0.997 (gray line). **f** Temperature-dependent PL spectra of ZnO nanocrystals (from 420 to 850 nm). **g** The FWHM of emission spectrum as a function of temperature, and the solid line is the fitting result. **h** A1 LO phonon spectra of ZnO nanocrystals. **i** 3D top-view TA spectra of the ZnO nanocrystals under 355 nm fs-laser excitation. **j** TA spectra of the ZnO nanocrystals at indicated time from −1 to 2800 ps under 355 nm fs-laser excitation.

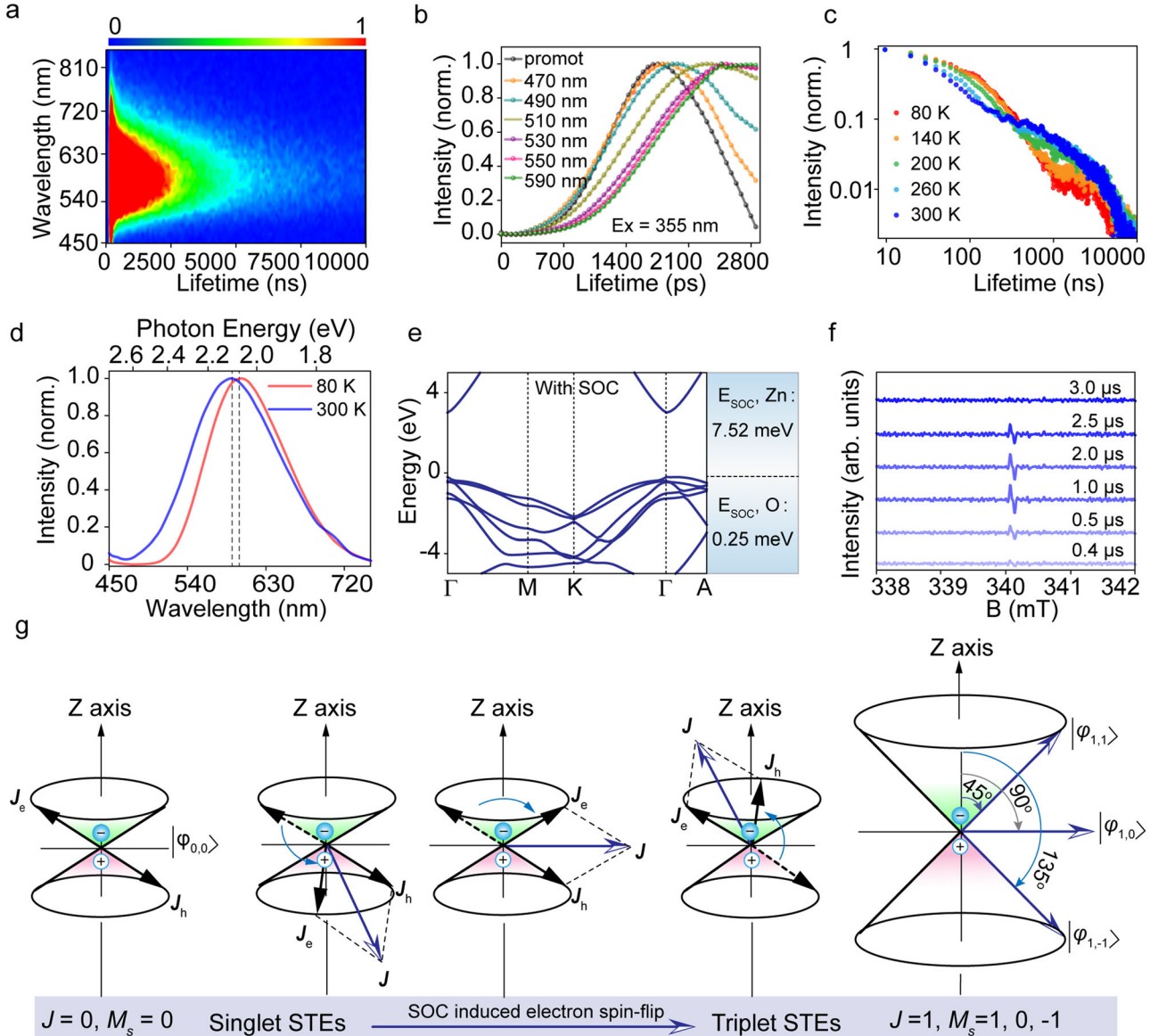

**Fig. 3 | Transient spectra and singlet/triplet mixed excitons configuration.**
**a** Contour plots of time-resolved emission spectra of the ZnO nanocrystals under 350 nm fs-laser excitation at room temperature. **b** PL decay curves of the ZnO nanocrystals at selected wavelength under 375 nm excitation at room temperature. **c** Temperature-dependent PL lifetime decay spectra. **d** The PL spectra of ZnO nanocrystals at 80 K and 300 K. **e** Orbit-resolved projected density of states of ZnO with SOC. **f** Time-resolved ESR spectra of ZnO nanocrystals after a 355 nm laser flash at room temperature. **g** Schematic diagram of the spin vector of singlet and triplet excitons (electron-hole pairs).

This suggests that the atomic distortion induced by charge transfer promotes the enhancement of the electron-phonon coupling. This enhancement will in turn promotes the expansion of the trapped charge. Transient absorption (TA) spectra further can be used as a useful tool to investigate self-trapping time of free excitons and free carriers. A broad excited state absorption can be observed from 3D TA spectra shown in Fig. 2i, indicating high excited state density of the ZnO nanocrystals. Moreover, no bleaching feature can be found in this wide range, meaning no sign of filling of permanent trap states[17]. The ZnO nanocrystals show a broad absorption from 490 to 770 nm, corresponding to absorption from STE to FE or FC. The detailed relaxation process of excited states versus time is shown in Fig. 2j. The STE absorption reaches the maximum value at around 300 fs, indicating all the STEs can be captured within 300 fs and an onset time of around 160 fs can be obtained, consistent with the exciton self-trapping time (115.3 fs) calculated by fitting the temperature-dependent FWHM[18]. The above results confirm that the bright

emission of the ZnO nanocrystals is attributed to the radiative recombination of STE states.

## Singlet/triplet mixed STE emission
Interestingly, two distinct lifetimes in the ZnO nanocrystals (Fig. 3a) are observed in 3D time-resolved emission spectra, indicating difference recombination processes are existed. The evolution and recombination process of the light-induced excited states of the ZnO nanocrystals was investigated, in order to further explore the emission mechanism. Nanosecond and microsecond lifetime across the probe region (from 400 nm to 850 nm), and the lifetime of the ZnO nanocrystals collected at 590 nm can be well fitted by bi-exponential function with lifetimes of $\tau_1 = 93$ ns (63%) and $\tau_2 = 1.53\,\mu s$ (37%) in Fig. 1d. The fast component is assigned to recombination of singlet STE, while the slow component can be assigned to recombination of triplet STE (electron and hole have parallel spins), which is discussed in the next. The PL onset time shows a wavelength-dependent property,

which the emitted photon with low-energy arises slowly compared with that of high-energy (Fig. 3b and Fig. S7), and this is direct evidence of STE. Similar emission-wavelength-dependent relaxation dynamics have been reported in perovskite $(C_6H_{11}NH_3)_2PbBr_4$ with STE emission property[37]. Temperature-dependent PL decay dynamics measurement provide clearly insight for identification of exciton state types[38]. Thus, the temperature-dependent PL-decay dynamics of ZnO nanocrystals from 80-300 K were recorded, as shown in Fig. 3c and Fig. S8a. The emission short lifetime ($\tau_1$, from 159 to 38 ns) decreases with increasing temperature. The long lifetime ($\tau_2$) gradually increases first (80–160 K) and then decreases (160–300 K). Such behavior can be attributed to two aspects: enhanced reverse intersystem crossing (RISC) process from long-lived triplet excited state back to short-lived singlet excited state and enhanced exciton-phonon interaction[39]. In cryogenic region (80–140 K), the long lifetime is suppressed compared with high temperature region (over 140 K), due to insufficient energy for electrons in triplet state back to singlet state[40]. Temperature-dependent average lifetimes ($\tau_{ave}$) of ZnO nanocrystals at 180–300 K are shown in Fig. S8b, and $\tau_{ave}$ versus temperature is fitted by using a thermally activated delayed fluorescence model[38]:

$$\tau_{ave} = \frac{3 + \exp(-\frac{\Delta E_{ST}}{K_B T})}{3/\tau_T + 1/\tau_S \exp(-\frac{\Delta E_{ST}}{K_B T})} \qquad (2)$$

where $k_B$ is the Boltzmann constant, $\Delta E_{ST}$ represents the singlet-triplet energy splitting, $\tau_S$ and $\tau_T$ are the single and triplet emission lifetime at 0 K, respectively. The fitted energy between singlet ($S$) and triplet ($T$) $\Delta E_{ST}$ is 60 meV, which also agrees with the measured energy ($\Delta E_{ST} = 1240/588-1240/601 = 46$ meV) from low temperature and room temperature spectra, as shown in Fig. 3d. The SOC energy of atoms zinc and oxygen are 7.52 and 0.25 meV, confirming the efficient SOC in this system. The singlet-triplet energy splitting was 58 meV for the ZnO nanocrystals obtained by the theoretical calculations, which were close to the experimental results, as shown in Fig. 3e. In addition, time-resolved electron spin resonance (ESR) as a powerful tool for investigation unpaired electrons was used to studied excited electrons in this work[41]. Time-resolved ESR spectra of the ZnO nanocrystals were recorded under after 355 nm laser flash within the detection time window of 0.4–3 μs at room temperature, as shown in Fig. 3f. An obvious polarized spin resonance signal can be observed and the intensity of this signal increases first and then decreases with time, which is indicative of unpaired electrons following electrons spin flip after excitation. Then, the flipped electron and hole formed triplet exciton through Coulomb interaction[42]. The ESR signal of the ZnO nanocrystals disappears after excitation for 2.5 μs, indicating the lifetime of triplet excitons is about 2.5 μs, which is agreement with long-lived component in emission lifetime. Until now, STEs in the ZnO nanocrystal includes two different excitons: singlet STEs and triplet STEs, all of them contribute to STE emission. When electron with angular momentum of 1/2 in ground state was excited to excited state, a hole with angular momentum of −1/2 was generated simultaneously. The total spin $J$ is 0 or 1 when their angular momentum is combined, and exciton splits to singlet and triplet exciton correspondingly[26]. The evolution process from singlet exciton to triplet exciton is illustrated in Fig. 3g. Spin-orbit coupling provides driving force for electrons to transition from singlet state to triplet state (ISC)[43]. Exciton is typical two particles system, and the corresponding wave functions of electron and hole are consisted of spatial wave functions and spin wave functions. If the spin-orbit coupling interaction was considered to construct triplet state wavefunctions, the wavefunctions can be described as: $\Phi = \varphi_{space} \times \varphi_{spin}$, where $\Phi$ is total wavefunctions, $\varphi_{space}$ is space wavefunction, $\varphi_{spin}$ is spin wavefunction. For construction of wavefunction $\Phi$, we should construct $\varphi_{space}$ and $\varphi_{spin}$ firstly. As we know that the valence band of ZnO are filled with electrons (ground

state), O 2$p$ and Zn 4$s$ states are taken as ground state and excited state for intuitive understanding. Thus, we can use 'two particles' model to deal with this issue. For the ground state, the total orbit angular $L$ is 2, 1, 0, and the total spin angular $S$ is 0. For $L$-$S$ coupling, the total angular $J$ is 2, 1, 0, and the detailed information is listed in Table S3. For the excited state, the total orbit angular $L$ is 1, and the total spin angular $S$ is 1, 0, and the detailed information is listed in Table S4. The overall wavefunction is always antisymmetric by construction, we can make any linear combinations of space and spin wavefunctions, and we still obtain an eigenstate. In addition, we should make sure that all the space and spin wavefunctions are individually normalized in order to construct the wavefunctions, as shown in Fig. S9. Because the Hamiltonian only depends on spatial variables and not spin, we can conclude the triplet states are degenerate states. Here, we just discuss spin wave functions in this work. Pauli exclusion principle states that two fermions in the same system do not occupy same quantum state. Thus, for $J = 0$ singlet exciton, the space wave functions are symmetric, thus the spin wave function must be antisymmetric:

$$|\varphi_{0,0}\rangle_{singlet} = \frac{1}{\sqrt{2}}[|\uparrow_e\rangle|\downarrow_h\rangle - |\downarrow_e\rangle|\uparrow_h\rangle] \qquad (3)$$

For $J = 1$ triplet exciton, the spin wave functions are symmetric, as follows:

$$|\varphi_{1,1}\rangle_{triplet} = |\uparrow_e\rangle|\uparrow_h\rangle \qquad (4)$$

$$|\varphi_{1,0}\rangle_{triplet} = \frac{1}{\sqrt{2}}[|\uparrow_e\rangle|\downarrow_h\rangle + |\downarrow_e\rangle|\uparrow_h\rangle] \qquad (5)$$

$$|\varphi_{1,-1}\rangle_{triplet} = |\uparrow_e\rangle|\uparrow_h\rangle \qquad (6)$$

where $|\varphi_{J,m_z}\rangle$ is spin wave functions, and $m_z$ is the $z$ projection of $J$. For radiative recombination of electron and hole, they must have anti-parallel spins, this process is fast. Or else, the coordinated process with different total spin angular momentums is unlikely occurred, thus triplet excitons are poorly emitting, and this process is slow if it occurs[20,26]. The schematic diagram of PL mechanism of the ZnO nanocrystals is shown in Fig. S10. Singlet STEs are generated when the ZnO nanocrystals absorb high energy photons, spin flip occurs in some of electrons due to SOC effect. The spin flipped electrons form triplet excitons by combining with parallel spin holes, the electrons in triplet state can return to singlet state through RISC process with assistance of temperature. Thus, singlet and triplet mixed excitons contribute to efficient emission of the ZnO nanocrystals, and the corresponding contribution ratio is 63% and 37%.

In addition to 0D nanoparticles, 3D bulk ZnO microwires were also explored their emission properties in visible region. Digital photos of ZnO microwires under the sunlight and UV light excitation were shown in the Fig. S11a. The SEM images demonstrated that ZnO microwires were uniform thickness with diameter of around 10 μm and length of over 100 μm (Fig. S11b and Fig. S11c). EDS mapping (Fig. S11d) result indicated the element composition the microwires. In addition, the TEM images (Fig. S11e and Fig. S11f) illustrated that the bulk ZnO shows clear lattice fringes, which indicated that the ZnO microwires had good crystallinity. These ZnO microwires exhibited single-phase and single-crystal wurtzite structure with a preferred (002) orientation without any other peaks based on the measured XRD result (Fig. S11g). Additionally, the PL excitation (PLE) spectrum shows that the optimal excitation peak is located at 375 nm, and the corresponding emission spectrum is composed of a weak band peaking at 390 nm (FE emission) and a strong band peaking at 520 nm (STE emission) (Fig. S11h). Thus, large Stokes shift (0.93 eV) and broad spectrum (full width at half-

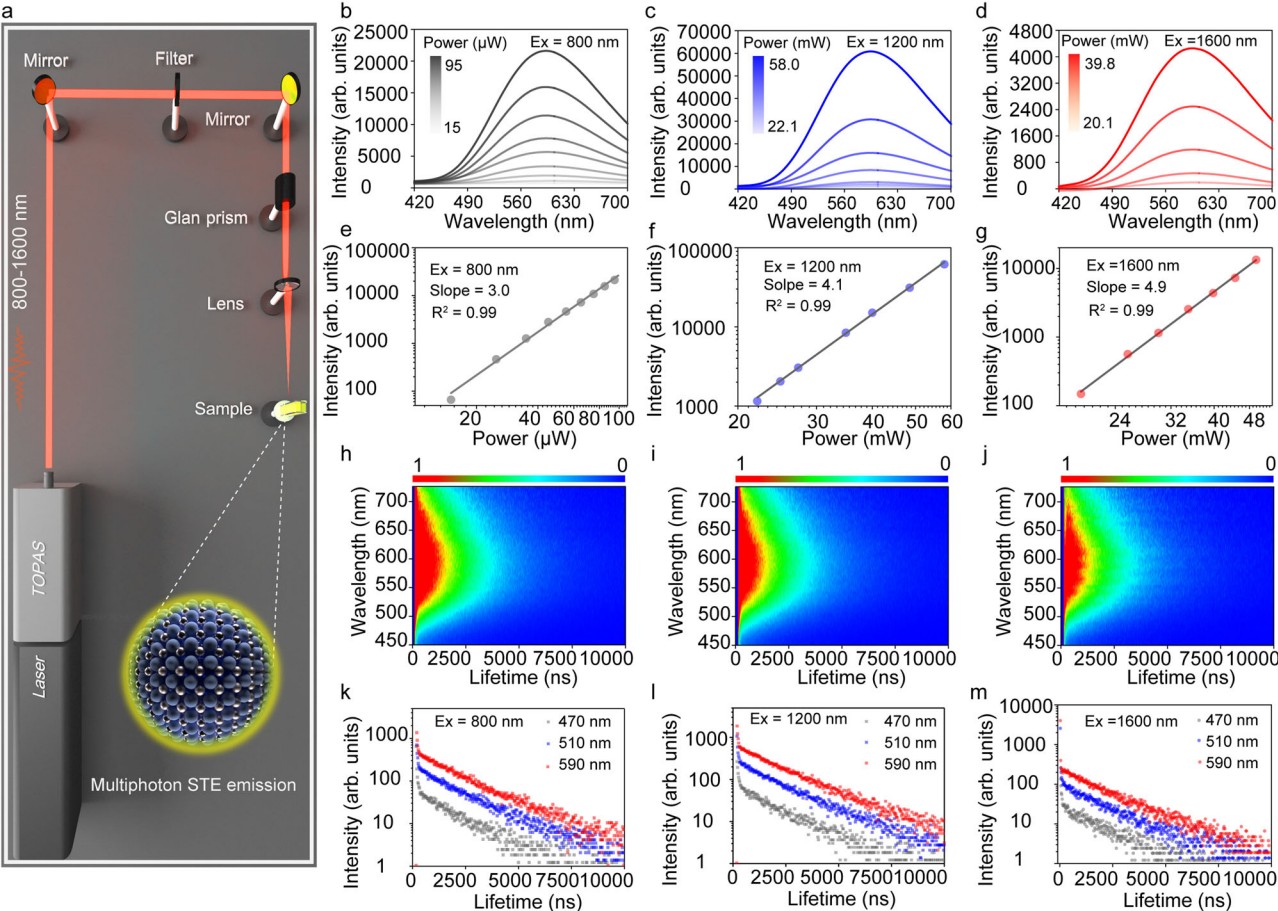

**Fig. 4 | Multiphoton excited singlet/triplet mixed STE emission. a** Schematic illustration of experimental setup of multiphoton excited emission measurement system based on a fs-laser. The wavelength can be adjusted from 800 nm to 1600 nm with the temporal width of 50 fs. **b** Emission spectra of the ZnO nanocrystals under 800 nm fs-laser excitation with different excitation intensities at room temperature. **c** Emission spectra of the nanocrystals under 1200 nm fs-laser excitation with different excitation intensities at room temperature. **d** Emission spectra of the nanocrystals under 1600 nm fs-laser excitation with different excitation intensities at room temperature. Cubic (**e**), quartic (**f**) and quintic (**g**) dependence of the emission intensity in ranges of 450–700 nm on excitation intensity, respectively. Time-resolved emission spectra of the nanocrystals under 800 nm (**h**), 1200 nm (**i**) and 1600 nm (**j**) fs-laser excitation at room temperature, respectively. PL decay curves of the nanocrystals at selected wavelength under 800 nm (**k**), 1200 nm (**l**) and 1600 nm (**m**) fs-laser excitation, respectively.

maximum (FWHM) of 140 nm, 0.66 eV) for the strong emission band of ZnO microwires were similar to emission of ZnO nanocrystals from STE. As shown in Fig. S11i, the identical shapes and features of excitation spectra suggest that the broad emission originates from the recombination of the same initial excited state. Huang-Rhys factor (*S*) of the bulk ZnO microwires is 17.3 by fitting temperature-dependent full width at half maximum (FWHM) from 300 to 80 K (Fig. S11j and Fig. S11k), which is lower than that of ZnO nanocrystals (30.8). The degree of fitting (R-squared) was 0.97, indicating the fitted result was reliable. The lower S value was attributed fact that ZnO nanocrystals were more prone to lattice distortion than bulk ZnO microwires. The ratio of FE emission in ZnO microwires and nanocrystals are 6.2% and 1.3%, indicating that more FEs were trapped to from STEs in nanocrystals. Previous reports also confirmed that STE formation was believed to have a strong relationship with the dimensionality of the system, therefore, the visible STE emission depended on nanostructure size and shape[19]. The phonon energy of the excited states was 48.29 meV, being close to the value of ZnO nanocrystals, and calculated STE binding energy of ZnO microwires is ~86.23 meV by fitting the curve of the relationship between temperature and integrated PL intensity (Fig. S11l). Furthermore, the broad emission from ZnO microwires versus excitation powder over 3 orders of magnitude shows linear dependence on the excitation intensity (Fig. S12a and S12b), indicating that the broad emission does not arise from permanent defects[19]. The lifetime of ZnO

microwires (520 nm) was also divided into short lifetime (15.44 ns, 97.5%) and long lifetime (108.32 ns, 2.5%) by fitting the PL decay curve with a double exponential function (Fig. S12c). Similar fast and slow component can also be observed, corresponding to the recombination of singlet STEs and triplet STEs, respectively. These results indicate that the broad PL emission from bulk and nanoscale ZnO originates from the radiative recombination of singlet/triplet mixed STE emission.

## Multiphoton properties and bioimaging applications

In view of the efficient emission of the ZnO nanocrystals and advantage of multiphoton excited emission in the field of deep tissue bioimaging, the emission property of the nanocrystals under multiphoton excitation was studied. Multiphoton excited band edge emission of ZnO has been reported before[6], while multiphoton excited STE emission has not been investigated. To measure the multiphoton excited emission of the ZnO nanocrystals, an experimental setup of measurement system based on femtosecond (fs) laser has been built, as illustrated in Fig. 4a. It has been reported that NIR window lights (NIR-I, 650–950 nm; NIR-II, 1100-1350 nm; NIR-III, 1600–1870 nm; NIR-IV, centered at 2200 nm) have unique advantages in deep tissue bioimaging. Thus, the emission properties of ZnO nanocrystals under the excitation of NIR window lights were investigated. The PL spectra of the ZnO nanocrystals under 800 nm (NIR-I window), 1200 nm (NIR-II window) and

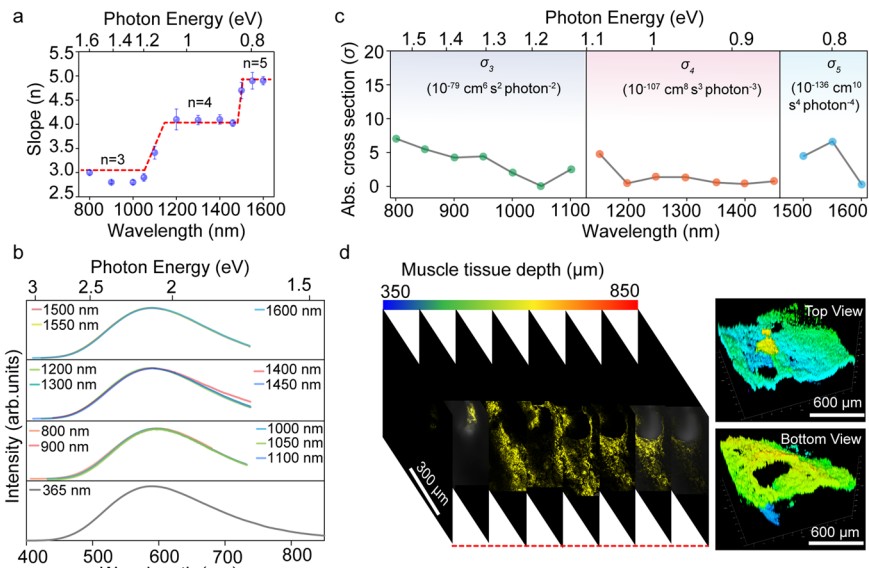

**Fig. 5 | Multiphoton properties and bioimaging applications of the ZnO nanocrystals. a** The slopes plotted as a function of laser excitation wavelengths. Error bars represent ±1 standard deviation from the mean, m = 3. **b** Emission spectra of the ZnO nanocrystals under different excitation wavelengths at room temperature. **c** Three-photon absorption cross-section ($\sigma_3$), four-photon absorption cross-sections ($\sigma_4$) and five-photon absorption cross-section ($\sigma_5$) of the ZnO nanocrystals in the wavelength range 800–1600 nm at room temperature. **d** Two-photon emission images of mice muscle tissue at different focal depths after injection of 200 μL 50 μg mL$^{-1}$ ZnO nanocrystals PBS solution at room temperature.

1600 nm (NIR-III window) were recorded, as shown in Fig. 4b–d. The corresponding slopes of the PL intensity versus excitation fluence are plotted in Fig. 4e–g. Remarkably, cubic, quartic, and quintic dependence of the emission intensity on excitation fluence under 800 nm, 1200 nm and 1600 nm can be observed, corresponding to three-, four- and five-photon excited emission[4]. The multiphoton excited PL spectra show identical emission wavelength (centered around 590 nm) with single photon excited PL spectra, indicating the emission origins from STE radiative recombination. Multiphoton excited STE emission of the ZnO nanocrystals in visible range will promote their potential application in multiphoton bioimaging. Specially, NIR-III window (1600–1870 nm) with larger penetration depth, which is also called golden window, has aroused the attention of researchers. In order to investigate emission origins under multiphoton excitation, 3D time-resolved emission spectra under excitation of 800–1600 nm were recorded, as shown in Fig. 4h–j. Obviously, two distinct lifetimes can be observed, and all of lifetimes show identical decay tendency under excitation of light with different wavelengths, indicating the emission origins from same excited states. In addition, the emission decay process under multiphoton excitation has similar behavior with single photon excited emission. Furthermore, PL decay spectra at selected wavelengths (470, 510 and 590 nm) have long lifetime at long wavelength due to the strong coupling between excitons and phonons (Fig. 4k–m). The lifetime of ZnO nanocrystals shows slight increase in wavelength range 470–590 nm gradually, consistent with the origin of STE emission. The distinct fast and slow component in lifetime decay spectra under excitation of multiphoton correspond to singlet excitons and triplet excitons radiative recombination. It is a plausible conclusion that multiphoton excited singlet/triplet mixed STE emission have been demonstrated in the ZnO nanocrystals.

The PL spectra of the ZnO nanocrystals under excitation of lights with different wavelengths are also measured and the slopes of the PL intensity versus excitation fluence are calculated, as shown in Figs. S13–S15 and Fig. 5a. The PL intensity of the ZnO nanocrystals increases with the excitation power under the excitation of 800–1600 nm. The slope was around 3.0 in the range of 800-1100 nm, clearly indicating three- photon excited STE emission. For excitation wavelength located at 1200–1450 nm, the slopes are around 4, indicating the dominance of

four-photon excited process. With the increase of excitation wavelength (1500–1600 nm), the slope increases to around 5, revealing a switch of the excitation mechanism to five-photon excited process. For clearly observation, one-photon, three-photon, four-photon and five-photon excited STE emission spectra of the ZnO nanocrystals under different excitation wavelengths are shown in Fig. 5b, and the coincident emission wavelength also indicates that the bright yellow emission under multiphoton excitation is similar to one-photon excited STE emission.

The multiphoton absorption cross-sections of ZnO nanocrystals were investigated seriously. The open-aperture Z-scan requires very high excitation peak intensity due to its low detection efficiency in the characterization of higher-order multiphoton absorption (such as four- photon and five-photon absorption)[4]. But the extremely high excitation intensities may result in sample damage or introduce other effects (or artifacts). Therefore, the multiphoton absorption (such as four-photon and five-photon absorption) were performed by combining multi-photon excited PL measurements with open-aperture Z-scan technology. open-aperture Z-scan has been performed at 800 nm in view of negligible absorption in this region (Figs. S16 and S17). The three-photon absorption cross section $\sigma_s$ at 800 nm were calculated as $0.5 \times 10^{-78}$ cm$^6$ s$^2$ photon$^{-2}$, $0.93 \times 10^{-78}$ cm$^6$ s$^2$ photon$^{-2}$ and $0.69 \times 10^{-78}$ cm$^6$ s$^2$ photon$^{-2}$, respectively. The average three-photon absorption cross section $\sigma_s$ at 800 nm was $0.71 \times 10^{-78}$ cm$^6$ s$^2$ photon$^{-2}$. Then, multiphoton absorption cross-sections of ZnO nanocrystals at other wavelengths were obtained by utilizing the three-photon absorption cross sections at 800 nm measured by Z-scan technique and multiphoton-excited PL technique (Fig. S18). Because the ethanol solvents exhibited varying degrees of absorption in 1150–1600 nm range (Fig. S16). The change of laser power after passing ethanol have been measured to exclude the effect of solvent (Fig. S19). The three-photon cross section ($\sigma_3$), four-photon cross section ($\sigma_4$) and five-photon cross section ($\sigma_5$) under different excitation wavelength are calculated by the formula 13, 14 and 15[44]. Thus, the three-photon cross-section ($\sigma_3$) are derived to be $0.02$–$0.7 \times 10^{-78}$ cm$^6$ s$^2$ photon$^{-2}$, as shown in the left of Fig. 5c and Table S5. Four-photon cross-section ($\sigma_4$) of the nanocrystals ($0.04$–$0.48 \times 10^{-106}$ cm$^8$ s$^3$ photon$^{-3}$) are shown in the middle of Fig. 5c and Table S6. Likewise, five-photon cross-section ($\sigma_5$) are

$0.33-6.47 \times 10^{-136}$ m$^{10}$ s$^4$ photon$^{-4}$ (the left of Fig. 5c and Table S7). The result demonstrates that the synthesized ZnO nanocrystals have excellent multiphoton excited emission property, which lays solid foundation for 3D deep tissue bioimaging[45]. Simultaneously, ZnO nanocrystals have low biological toxicity and good photostability (Fig. S20). Thus, the nanocrystals have been successfully used for two-photon imaging of B16-F10 cells (Fig. S21), which is clearer than one-photon imaging because of low background fluorescence interference[46]. In addition, the histological analysis reveals that no noticeable damage or inflammatory lesions are observed in major organs (brain, heart, lung, liver, spleen, and kidney) of the mice after treating with the ZnO nanocrystals for 7 days (Fig. S22a), suggesting that the nanocrystals have good biocompatibility and are expected to be used in vivo 3D bioimaging[47]. Next, the laser confocal system with femtosecond laser is used to collect imaging signals at regular 50 μm intervals (Fig. S22b and S22c). Fig 5d and Fig. S22d show that the two-photon imaging signal appears in the range of 350-850 μm from the surface of the muscle tissue. According to two-photon imaging of muscle tissue, the three-dimensional structure of the tissue from different perspectives (such as top view and bottom view) was clearly constructed. The 850 μm imaging depth in vivo is at a high level compared to other materials (Table S8). According to two-photon imaging of muscle tissue, the three-dimensional structure of the tissue from different perspectives (such as right view and left view) was also clearly constructed. Carbon nanodots as an efficient ecofriendly multiphoton imaging agents have been investigated for multiphoton bioimaging, we chose two kinds of efficient red emission carbon dots (CNDs-1 and CNDs-2) with PL QYs of 20% and 57% to evaluate multiphoton bioimaging ability objectively. The excitation and emission spectra of CNDs-1 and CNDs-2 were measured and shown in Fig. S23. In total, 200 μL CNDs-1 (100 μg mL$^{-1}$) and CNDs-2 (50 μg mL$^{-1}$) PBS solution were injected into the mice muscles. Then the imaging depth of CNDs-1and CNDs-2 in vivo was evaluated to 450 μm and 1010 μm, respectively (Fig. S24). Thus, the 850 μm imaging depth of ZnO nanocrystals in vivo was at a high level compared to other materials.

## Discussion

ZnO with emission in visible region has a very long and complex history, and the broadband visible emissions have been found in ZnO with varied sizes and morphologies (ZnO nanocrystals, ZnO nano-cones and nano-rods, ZnO films and ZnO microwires). Previous works attributed the visible emission to the presence of impurities and structural defects. But we demonstrated that the broad emission in ZnO nanocrystals with high PL QY 60.5% attributed to singlet/triplet mixed STE emission. A large full width at half-maximum (617 meV) and Stokes shift (1.29 eV), PL depending linearly on excitation intensity over 3 orders of magnitude and 160 fs onset time confirm the presence of STE emission. Temperature dependent steady state, transient state and time-resolved ESR spectra clearly showed that the STEs contained singlet and triplet STEs, which contributed to the high PL QY. First-principles calculations with the state-of-the-art hybrid functionals and algorithms of ZnO crystal upon photoexcitation also confirmed the nature of STE emission in ZnO nanocrystals.

Furthermore, the visible emission characteristics of bulk ZnO were studied and discussed. The Huang-Rhys factor (S) of the bulk ZnO microwires (17.3) was lower than that of ZnO nanocrystals (30.8), indicating that ZnO nanocrystals were more prone to lattice distortion than bulk ZnO microwires. The ratio of FE emission in ZnO microwires and nanocrystals are 6.2% and 1.3%, demonstrating that more FEs were trapped to from STEs in nanocrystals. The visible emission branching ratio depended on nanostructure size, indicating STE formation has a strong relationship with the dimensionality of the system. In other words, a large electron-phonon coupling was crucial to the formation of STE, and the low crystal dimensionality was favor of distortion and STE formation, as well as high emission efficiency because of the strong quantum confinement particularly in the 0D crystal structure. As discussed before, a reduced dimensionality and localized electrons and holes contribute to efficient STE emission.

Subsequently, the multiphoton properties of ZnO nanocrystals were evaluated, the nanocrystals can be excited effectively under of 800-1600 nm light. Based on the outstanding PL stability, high PL QY, good water dispersibility and biocompatibility, ZnO nanocrystals as a fluorescent probe have been applied for multiphoton bioimaging with a depth of up to 850 μm.

## Methods
### Materials
All the chemicals were analytical grade without further purification: Zinc aetate dihydrate (Zn(Ac)$_2$ • 2H$_2$O), potassium hydroxide (KOH), (3-aminopropyl) triethoxysilane (APTES), Phosphate buffer solution (PBS) were purchased from Aladdin Chemistry Co. Ltd. (Shanghai, China). Absolute ethanol was obtained from Tianjin Yongda Chemical Reagent Co. Ltd. (Tianjin, China).

### Characterization
The X-ray diffraction (XRD) was recorded by X' Pert Pro diffractometer with Cu Kα radiation (D8 Discover, Bruker, Germany). The morphology of the sample is characterized by transmission electron microscopy (TEM) (HT7700, Hitachi, Japan). The emission spectra of the ZnO nanocrystals were analyzed by a fluorescence spectrophotometer (F-7000, Hitachi, Japan; FLS-1000, Edinburgh Instruments, UK). The UV-Vis absorption of the ZnO nanocrystals was performed by a spectrophotometer (UH4150, Hitachi, Japan). The hydrodynamic diameter distribution of the sample was acquired by a malvern zetasizer nano series (ZEN5600, Malvern, UK). FTIR spectra of the nanocrystals were measured by a spectrometer (Nicolet 6700, Thermo Fisher Scientific, USA). The XPS measurements were obtained by a spectrometer (EscaLab 250Xi, Thermo Fisher Scientific, USA). The PL decay were determined by a time-corrected single photon counter system (FLS-1000, Edinburgh Instruments, UK). The PL QY of the sample was measured by a calibrated integrating sphere of a spectrometer (FLS-1000, Edinburgh Instruments, UK). The cellular and mice muscle tissue imaging pictures were acquired at a laser scanning confocal microscope (Leica TCS SP8 STED 3X, Leica Microsystems, Germany).

### Synthesis of ZnO nanocrystals
Firstly, 5.5 g Zn(Ac)$_2$ • 2H$_2$O and 2.0 g KOH were dispersed in 150 mL and 20 mL absolute ethanol, respectively. Then, the KOH solution was added into the Zn(Ac)$_2$ • 2H$_2$O solution followed by continuous stirring for 4 h. In total, 500 μL APTES was added into the above complex solution under continuous stirring for 3 h. The precipitate was centrifuged and washed by absolute ethanol for three times. Finally, ZnO nanocrystals were obtained after drying in a blast oven at 70 °C for 5 h.

### PL QY measurement
The absolute PL QY of the ZnO nanocrystals was measured by a FLS-1000 spectrometer with calibrated integrating sphere. The average PL QY of the nanocrystals is about 60.5% in the range of 400–800 nm after multiple measurements.

### Cell culture
Hela and B16-F10 cells were purchased from the Chinese Academy of Sciences Cell Bank. These cancer cells were incubated in Roswell Park Memorial Institute (RPMI) 1640 medium with 10% fetal calf serum (FBS) and 1% penicillin streptomycin, respectively[48]. Then the cultures were placed in an incubator at 37 °C with 5% CO$_2$ and 95% air.

### Cytotoxicity measurements
Hela cells were used to evaluate the cytotoxicity of the ZnO nanocrystals by CKK-8 assay[49]. The cancer cells were cultured in 1640

medium with 10% fetal calf serum and 1% penicillin streptomycin and placed on 96-well plates at 37 °C with 5% $CO_2$ and 95% air for 24 h. Then, ZnO nanocrystals with different concentrations were added into each well and the plates were incubated for another 24 h. In total, 10 μL CCK-8 solution was added to the plates for 2 h. The absorbance of the sample was performed by a microplate reader (BIO-RAD 550, BIO-RAD Laboratories, USA). The cell viabilities (%) were calculated by the following formula:

$$Cell\ viability\ (\%) = Abs_{sample}/Abs_{reference} \qquad (7)$$

where $Abs_{sample}$ was measured in the absence of the ZnO nanocrystals, and $Abs_{reference}$ was performed in the presence of the nanocrystals, respectively.

### Animal experiments
Female mice (aged 8 weeks) were obtained from Vital River Laboratory Animal Technology Co. Ltd (Beijing, China). In all the animal experiments, mice were kept in a pathogen-free facility. All animal experiments were performed in accordance with the Zhengzhou University Guide for Care and Use of Laboratory Animals, and the procedures received approval from the Animal Ethics Committee of Zhengzhou University. The animal laboratory's accreditation number is SCXK (YU) 2019-0004.

### Histological study
Eight-week-old mice were randomly divided into two groups of four mice each. Different treatments were applied prior to the measurement by group 1 and group 2. In total, 20 μL PBS was injected into group 1 as the control. Mice in group 2 were treated with 200 μL ZnO nanocrystals (50 μg mL⁻¹) in PBS. These two groups of mice were sacrificed 7 days after injection to gain insight into the acute and long-term toxicity. The tissues from the brain, heart, lungs, liver, spleen, and kidneys were harvested and fixed in 4% paraformaldehyde. After embedding in paraffin, sections of the samples were cut, stained with H&E, and then viewed under an optical microscope.

### Imaging of cancer cells in vitro
B16-F10 cells were cultured in RMPI 1640 medium containing 10% FBS and 1% antibiotics. ZnO nanocrystals are dispersed in DMSO. Subsequently, 1 mL of the suspension was added to the culture medium, and then the cells were placed in an incubator at 37 °C in 5% $CO_2$ for 12 h. The cells were washed for three times with warm phosphate buffer to remove excess the nanocrystals. Then, the cells were fixed on a glass slide to take pictures of one-photon and two-photon cell imaging using a Leica TCS SP8 STED 3X laser scanning confocal microscope.

### In vivo two-photon imaging
After the mice were anesthetized, 200 μL ZnO nanocrystals (50 μg mL⁻¹) PBS solution was injected into the mice muscles. The muscles treated with the Nanocrystals were collected for two-photon imaging in vivo by using a Leica TCS SP8 STED microscope. Multiphoton images of muscle tissue were taken every 50 μm interval to realize 3D construction of muscle tissue through software.

### Transient absorption spectroscopy
The transient absorption spectrum of ZnO nanocrystals was acquired by a regenerative amplified femtosecond Ti: Sapphire laser system (Coherent Legend Elite HE + USP-1K − III, 35 fs, 1 kHz). Briefly, a 50% beam splitter was used to split the 800 nm output pulse into two parts. One part was used to generate a 350 nm pump laser pulse through an automated TOPAS Optical Parametric Amplifier (OPA). A synchronized chopper was used to chop the pump pulses to 500 Hz. The other part

was applied to generate a white light continuum (WLC) as a probe beam. Then, the probe beam was focused and passed through the sample by an Al parabolic reflector. After passing through the sample, the transmission changes of the probe light were detected by a fiber spectrometer (AvaSpec-ULS2048CL-EVO, Avantes). And the absorbance change of the sample was calculated with two adjacent probe pulses. 1 mg mL⁻¹ ZnO nanocrystals solution was added in a 2 mm optical length quartz cuvette to measure the transient absorption spectrum.

### Measurement of multi-photon excitation emission spectrum
Multiphoton singlet/triplet mixed STE emission spectra were performed by a Ti: sapphire laser system (Coherent Legend Elite HE + USP-1K − III, 35 fs, 1 kHz). The 800 nm beam was focused to an automated OPA to generate a wavelength-tunable laser pulse from 800 and 1600 nm as multiphoton excitation light source. 1 mg mL⁻¹ ZnO nanocrystals solution was placed in a 1 cm optical length quartz cuvette. And the multiphoton STE emission signal was collected at the left side of the quartz cuvette by a spectrometer (HRS-500 MS, PI) coupled with a liquid-nitrogen-cooled CCD (PyLoN, PI). In addition, a long-pass filter was used to filter out the scattered light of the excitation light source.

### Open-aperture Z-scan measurements for quantifying 3PA cross-sections ($\sigma_s$) at 800 nm
Open-aperture Z-scan measurements were conducted to quantify the $\sigma_s$ values (at 800 nm) of ZnO nanocrystals, which were dispersed in ethanol solvent and contained in 1-mm-thick quartz cuvettes. In open-aperture Z-scans, the excitation laser pulses (-100 fs, 800 nm, 1 kHz) were generated by a Ti: sapphire laser system (Coherent Legend Elite HE + USP-1K − III). A beam splitter was employed to divide the incident laser beam into two parts. The first part served as the reference and was directed into a reference power detector ($D_R$). The other part functioned as the signal beam and was focused onto a 1-mm-thick quartz cuvette filled with the ethanol solution of ZnO nanocrystals. The transmitted signal beam through ZnO nanocrystals was detected by a signal power detector ($D_S$). The sample transverses back and forth along the propagation direction of the laser beam (z-axis) on a linear motorized stage. The transmission of the sample ($D_S/D_R$) was monitored while translating the sample through the focal point, and the transmission was recorded as a function of the sample position (z). With the incident laser pulse energies kept at a constant level, the sample experiences various laser irradiance I(z) at different z-positions, giving rise to corresponding changes in transmission if the sample absorbs light nonlinearly. Three-photon cross-sections of the nanocrystals can be obtained through fitting the normalized transmittance to the well-established Z-scan theory[50] as:

$$z_0 = \pi\omega_0^2/\lambda \qquad (8)$$

$$L'_{eff} = [1 - \exp(-2\alpha_0 L)]/2\alpha_0 \qquad (9)$$

$$I_0 = I_{00}/(1 + z^2/z_0^2)^2 \qquad (10)$$

$$T(z) = 1 - \alpha_3 I_0^2 L'_{eff}/3^{\frac{3}{2}}(1 + z^2/z_0^2)^2 \qquad (11)$$

where $L'_{eff}$ is the effective sample lengths for three-photon absorption processes; $L$ is the sample length; $I_O$ is the excitation intensity at position $z$, $I_{OO}$ is the excitation intensity at position $z_0$; $z_0$ is the Rayleigh length; $\omega_0$ is the minimum beam waist at the focal point (0.26 cm) (z = 0); $\lambda$ is the laser free-space wavelength; $\alpha_O$ is the linear absorption coefficient, $\alpha_3$ is the three-photon absorption coefficient; $T(z)$ is the normalized energy transmittance at position z.

## Measurement of multiphoton absorption cross section by combining Z-scan and multi-photon excited PL technique

The open-aperture Z-scan requires very high excitation peak intensity due to its low detection efficiency in the characterization of higher-order multiphoton absorption (such as four- photon and five-photon absorption). But the extremely high excitation intensities may result in sample damage or introduce other effects (or artifacts). Therefore, most of the characterizations for the high-order multiphoton absorption (such as four-photon and five-photon absorption) were performed with multi-photon excited PL measurements and open-aperture Z-scan has been seldom used in this region. Therefore, in our manuscript open-aperture Z-scan has performed at 800 nm in view of negligible absorption in this region, and multiphoton absorption cross-sections of ZnO nanocrystals at other wavelengths were obtained by utilizing the three-photon absorption cross section $\sigma_s$ at 800 nm measured by Z-scan as the standard and combining with multiphoton excited PL technique. The pulse width and repetition frequency of the laser are 100 fs and 1000 Hz, respectively. $I_s$ was the photon density per unit time (photon $s^{-1}$ $cm^{-2}$). $I_O$ was the photon density per unit time (photon $s^{-1}$ $cm^{-2}$) at different wavelengths. Thus, $I_O$ was determined by the following equation:

$$I_0 = a^2 P / (\pi * 10^{-13} * 10^3 * 0.61^2 * 2^2 * \lambda * 10^{-7} * f^2 hc * 10^2)$$
$$= 2.14 * 10^9 \, a^2 P / \lambda f^2 hc \tag{12}$$

where $a$ was the incident spot diameter, $P$ was the laser power, $\lambda$ was the excitation wavelength (800–1600 nm), $f$ was the focal length (about 10 cm), $h$ was the Planck constant ($4.14 \times 10^{-15}$ eV s), $c$ was the speed of light ($3.0 \times 10^8$ m $s^{-1}$). Utilizing the three-photon absorption cross section $\sigma_s$ at 800 nm measured by Z-scan as the standard, the three-photon cross section ($\sigma_3$) of the nanocrystals excited by other wavelengths are obtained by the following formulas[51]:

$$F_3 / Fs = I_3^3 \sigma_3 / I_s^3 \sigma_s \tag{13}$$

where $F_3$, $I_3$ and $\sigma_3$ were the integrated intensity of the three-photon emission spectrum, the photon density per unit time and the three-photon cross section of ZnO nanocrystals under different excitation wavelength (850, 900, 1000, 1050 and 1100 nm), respectively. The four-photon ($\sigma_4$) and five-photon cross section ($\sigma_5$) of the sample were determined by these following equations[51]:

$$F_4 / Fs = I_4^4 \sigma_4 / I_s^3 \sigma_s \tag{14}$$

$$F_5 / Fs = I_5^5 \sigma_5 / I_s^3 \sigma_s \tag{15}$$

where $F_4$, $I_4$ and $\sigma_4$ were the integrated intensity of the four-photon emission spectrum, the photon density per unit time and the four-photon cross section of ZnO nanocrystals under different excitation wavelength (1150, 1200, 1300, 1400 and 1450 nm), respectively. $F_5$, $I_5$ and $\sigma_5$ were the integrated intensity of the five-photon emission spectrum, the photon density per unit time and the five-photon cross section of ZnO nanocrystals under different excitation wavelength (1500, 1550 and 1600 nm), respectively.

## Details of first-principles density functional theory calculations, Density functional perturbation theory calculations and ab initio molecular dynamics simulation

Our first-principles density functional theory calculations and ab initio molecular dynamics simulation are performed using the VASP code which is based on projector augmented wave (PAW) method[52,53]. The norm-conserving pseudopotentials with hybrid functional PBE0 functional[54] is employed to describe the interactions between the electrons and nuclear which has been proved to be accurate in

describing the band gap and lattice parameters of ZnO[55,56]. A $8 \times 8 \times 4$ k-mesh and kinetic energy cutoff of 800 eV are adopted and the atomic structure is fully relaxed until the forces acted on atoms are less than $10^{-5}$ $eV^{-1}$A. Spin-orbit interactions are omitted in potential energy surface calculation and ab initio molecular simulation while it is incorporated in the band structure and total energy calculation. The potential energy surface and molecular dynamics simulation of excited state are calculated using the constrained-DFT approach which has been demonstrated to be effective in handling ultrafast optical pumping phenomena[57,58].

## Data availability

All data generated in this study are provided in the Source Data file. Source data are provided with this paper.

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

## Acknowledgements

This work was supported by the National Natural Science Foundation of China (Grant Nos. 62075198, 11974317, 12274378), Outstanding Youth Foundation of Henan (Grant Nos. 222300420087).

## Author contributions

R.Z. and L.Z.S. contributed equally to this work. R.Z. synthesized, characterized and grew the nanocrystals and wrote the manuscript. L.Z.S. characterized the multiphoton excited emission spectra and the multiphoton absorption cross section. X.B.L. made theoretical calculations. K.K.L. conceived the experiments and wrote the manuscript. D.Y.G. wrote the manuscript. W.B.Z., S.Y.S. and C.F.L. characterized the nanocrystals. S.C. characterized the multiphoton absorption cross section. T.C.J. and Z.C. measured the biological toxicity of the sample. S.M. made theoretical calculations. C.X.S. supervised and coordinated all aspects of the project. All authors contributed to the discussion, writing, and

editing of the manuscript. The final version of the manuscript has been approved by all authors.

## Competing interests

The authors declare no competing interests.

## Additional information

**Peer review information** : *Nature Communications* thanks the other anonymous reviewer(s) for their contribution to the peer review of this work. Peer review reports are available.

