## [Peer review file · Nature Communications]

REVIEWER COMMENTS

Reviewer #1 (Remarks to the Author):

Referee's report on Nature Communications manuscript NCOMMS-22-41988

The authors report the results from an extensive series of measurements of the optical properties of ZnO nanocrystals. The nanocrystals have been synthesised using a chemical solution technique, and PL emission measurements, PL decay, transient absorption, nonlinear excitation of PL, and transient ESR measurements have all been performed on the nanocrystals (as well as XRD, HR-STEM, AFM, XPS etc.) The measurements performed constitute a very thorough body of characterisation work on the ZnO nanocrystals. Furthermore, the application of the nanocrystals in a bio-imaging application have also been explored and the results reported.

The execution of the research appears to be both painstaking and reliable. A detailed discussion of the nature and origins of the optical properties is given, in terms of the self-trapped exciton model. The optical characteristics of the emission seen from the nanocrystals is notable in a number of respects, including the high quantum efficiencies observed, and the long lifetimes. For these reasons undoubtedly the work reported will be of interest to the community of researchers studying the optical properties of ultra-small ZnO nanocrystals. Discussion of the origin of the visible emission band(s) in ZnO has a long and complex history, and the self-trapped exciton model (as opposed to e.g. oxygen vacancy defects etc.) is a relatively underexplored one. In the context of the present manuscript, while I don't feel that a crucial "clinching" argument is provided, I do think that the authors make a broadly convincing case that the STE model is a good candidate for the observed emission in these structures. Disentangling the various contributions to broadband visible PL from ZnO and definitively determining the origin of these has proven notoriously challenging. Green, red and other bands have been observed by many authors and with no widespread agreement as to the origin of these bands. It would have been interesting if the authors had discussed the relationship of the emission observed in their nanocrystals to these bands, typically observed in more "bulk-like" ZnO crystals (i.e. with crystal sizes much greater than the exciton Bohr radius of ~ 2.3 nm). However, I am quite sure that the visible emission observed in the present nanocrystals occurs at longer wavelengths than the "traditional" unstructured green luminescence band in ZnO, and is likely due to a different source. How it might relate to the many other observed visible bands is less clear, but the long PL lifetimes and very high quantum efficiency would seem to distinguish it from many of these.

My current review and comments are informed both by the present manuscript submitted to Nature Communications, as well as the response to my previous review of the original manuscript submitted to Nature Photonics, as well as consideration of the authors' response to some of the points made by the other referee in their review of the original manuscript submitted to Nature Photonics.

Based on the entirety of my review I would recommend that the present manuscript needs to be revised based on the comments below.

Overall scientific comments:

1. One issue which I thought about more and more during my reading of the revised manuscript, and which links back to both some parts of my comments on the original manuscript, as well as those of the other referee on the original manuscript, is in terms of the identification of the origin of the visible PL emission which is the subject of the present work. Specifically, the authors are adamant that the emission is intrinsic emission, originating from the self-trapping of a free exciton, rather than emission associated with a defect. The first question then must be, why is this strong visible emission not seen in all ZnO samples? There are plenty of reports of ZnO room temperature PL with dominant near UV FE emission, and little or no broadband visible emission. This includes emission from nanostructures of various sizes and morphologies. The reasons for such variation in an intrinsic emission process must be discussed. Does the branching ratio depend on nanostructure size, shape, surface termination etc.? The latter point also relates back to my query below on surface effects. The authors state that the absence of evidence in their electron microscopy support the identification as an intrinsic emission, but absence of evidence is not evidence of absence. And electron microscopy is not ideally suited to identifying point defects, because (a) the images are formed by passing through a finite volume of material, so the signal is an average along the path through the material, and also (b) since only a small sample of material is generally studied, which is not necessarily representative when looks for low concentrations of point defects. The authors also say that the linearity of the dependence of emission intensity on excitation power also supports the identification as an intrinsic emission, and that defect-related emission shows a sublinear effect effectively due to saturation of defect emission. However, the extent of the sublinearity is dependent on the degree of saturation, which depends on the local fluence and a number of other parameters. Ultimately the dependence of emission intensity on excitation power for defect emission is very often extremely close to linear, and essentially indistinguishable from linear behaviour. So I don't think the identification of the visible emission is as strong as the authors suggest. See also point 5 below under detailed comments. It is worth pointing out that many of the features claimed for the emission in this work can be shown by defect-related emissions; for example a bound exciton could possibly locally distort its lattice environment and trap itself even more deeply at a defect site with a large value of Huang-Rhys factor and broad emission band, and the singlet triplet behaviour is also possible for defect-related emissions. A self-trapping process of a bound exciton could also be strongly dependent on the overall nanocrystal size and morphology, and might also affect (reduce) the interaction with the surface. I am not saying I completely disagree with the authors' identification, but rather that I think other options need to be carefully considered in the context of the results in this work and also in the context of the wider literature relating to ZnO PL.

2. Related to the point above, and the one in my opening paragraphs, the authors should discuss the relationship of the emission observed in their nanocrystals to the other visible emission bands bands

typically observed in more “bulk-like” ZnO crystals (i.e. with crystal sizes much greater than the exciton Bohr radius of ~2.3 nm).

Overall presentation comments:

1. While the manuscript is generally quite well written and improved from the original version, some further editing and polishing would help.

Detailed comments:

1. A point I had mentioned in my first review was the use of the term “water solubility” or the like, when what is meant is “dispersability in water”. This issue still appears at lines 104, 189 and 467.

2. The very high PL QY of the synthesised nanocrystals in this study is one of the most notable features of the work. The lack of a degrading influence of the surfaces of the nanocrystals is particularly notable in this regard. I would like the authors to discuss the physical mechanisms which might contribute to such an unusual situation. I note that in the response to the referees’ comments on the original manuscript that the authors mentioned that ball milling had no noticeable effect on the PL properties, but the reasons why this might be the case need to be discussed. The authors also mention a silica coating can passivate surface defects. I asked about this latter point in my first review and the authors again said that ball milling had no effect and that the effect of surface states on emission had been eliminated. But it is completely unclear to me where a silica coating was applied to the present nanocrystals. When I look at the nanocrystal synthesis information there is no mention of silica encapsulation. And also, the XPS data in figure S2(f) doesn’t seem to show Si emission. This point needs to be discussed in better detail. For example, do the authors have any idea as to the nature of the ZnO nanocrystal surfaces (e.g. oxygen- or zinc-rich)?

3. In my first review of the original manuscript I mentioned the nature of the ZnO valence band edge states and that they form from spin-orbit coupled atomic states. I also asked that this be dealt with in more detail in the revised manuscript. This is important because the spin orbit coupling process is indicated as the original of triplet to singlet conversion. The authors do provide some discussion of this in the response document to the original comments, but ultimately they revert to treating the hole as a simple spin $\frac{1}{2}$ particle. However it is more correctly described as a doublet split off from a $J = 3/2$ spin orbit coupled manifold and this means that some of the effects of the spin orbit coupling interactions are built into the correct wavefunctions. I think the authors need to revisit this section of the manuscript.

4. The authors do address my comment in my review of the original manuscript concerning the notable peak in the 4 photon absorption cross-section at ~ 1150 nm, which they say is due to a higher single photon absorption cross-section at 287.5 nm, combined with the relatively high absorption of ethanol

solvent. They also show a figure, but to my mind, the data in the figure for the single photon absorption cross-section is quite continuous and doesn't show evidence of any peak (acknowledging that the y-axis is a logarithmic scale). Is it possible to utilise a different solvent which doesn't have an overlapping absorption onset?

5. The PLE data in figure 2(b) are interesting, in that wavelengths shorter than ~ 360 nm are very inefficient in exciting the visible emission band. This is quite different to the emission of the ZnO microwires in figure S10(h). If the emission is an intrinsic band, and not defect-related, then why should the PLE vary in this way?

Reviewer #2 (Remarks to the Author):

In the revised manuscript, the authors have well addressed most of my questions. The manuscript may be suitable for publication in nature communications. However, I still have several questions to be answered by the authors.

1. The authors claimed that singlet/triplet mixed exciton emission was responsible for the high emission efficiency of visible emission and the split energy between singlet and triplet exciton of 58 meV. It is well known that small split energy will be favorable for the occurrence of RISC. In addition, 58 meV is even much larger than the electron energy at 300 K (approximately 26 meV). In their case, How do electrons overcome this energy difference? The authors must well address this question and give the discussions in the text.

2. The authors claimed "it is not surprising that 3PA cross-sections of ZnO nanocrystals are much smaller than that of MAPbBr₃, while 4PA and 5PA cross-sections are comparable with MAPbBr₃." However, I'm still not satisfied with this response. Actually, the single intermediate state (SIS) approximation can be used to give an order-of-magnitude estimation (C. Xu and W. W. Webb, Topics in Fluorescence Spectroscopy, Vol. 5 (Springer, 1997), Chap. 11). The results of the authors deviate significantly from the predicted values. I strongly suggested the authors use other technique to determine the multiphoton absorption cross sections, such as Z-scan technique.

3. For the sudden maximum at 1150 nm in Figure 5c, the authors claimed it was caused by the higher single photon absorption at 287.5 nm and the relatively high absorption of ethanol solvent. I'm still not convinced by this explanation. It is obvious from the data given by the authors that the profiles of multiphoton and linear absorption spectra are quite different. In addition, the authors clearly knew that there is a significant absorption of solvent, why not try to eliminate this effect?

Response to Reviewers' comments

Dear Editors,

We are grateful for your consideration of this manuscript, and we also appreciate your suggestions. We sincerely thank the reviewers again for their insightful and constructive comments, which have been very helpful in improving the manuscript. The concerned issues raised by the reviewers are taken seriously, and a point-to-point response is listed as follows.

Reviewer #1 (Remarks to the Author):

Referee's report on Nature Communications manuscript NCOMMS-22-41988

The authors report the results from an extensive series of measurements of the optical properties of ZnO nanocrystals. The nanocrystals have been synthesized using a chemical solution technique, and PL emission measurements, PL decay, transient absorption, nonlinear excitation of PL, and transient ESR measurements have all been performed on the nanocrystals (as well as XRD, HR-STEM, AFM, XPS etc.) The measurements performed constitute a very thorough body of characterisation work on the ZnO nanocrystals. Furthermore, the application of the nanocrystals in a bio-imaging application have also been explored and the results reported.

The execution of the research appears to be both painstaking and reliable. A detailed discussion of the nature and origins of the optical properties is given, in terms of the self-trapped exciton model. The optical characteristics of the emission seen from the nanocrystals is notable in a number of respects, including the high quantum efficiencies observed, and the long lifetimes. For these reasons undoubtedly the work reported will be of interest to the community of researchers studying the optical properties of ultra-small ZnO nanocrystals. Discussion of the origin of the visible emission band(s) in ZnO has a long and complex history, and the self-trapped exciton model (as opposed to e.g. oxygen vacancy defects etc.) is a relatively underexplored one. In the context of the present manuscript, while I don't feel that a crucial "clinching" argument is provided, I do think that the authors make a broadly convincing case that the STE model is a good

candidate for the observed emission in these structures. Disentangling the various contributions to broadband visible PL from ZnO and definitively determining the origin of these has proven notoriously challenging. Green, red and other bands have been observed by many authors and with no widespread agreement as to the origin of these bands. It would have been interesting if the authors had discussed the relationship of the emission observed in their nanocrystals to these bands, typically observed in more “bulk-like” ZnO crystals (i.e. with crystal sizes much greater than the exciton Bohr radius of ~ 2.3 nm). However, I am quite sure that the visible emission observed in the present nanocrystals occurs at longer wavelengths than the “traditional” unstructured green luminescence band in ZnO, and is likely due to a different source. How it might relate to the many other observed visible bands is less clear, but the long PL lifetimes and very high quantum efficiency would seem to distinguish it from many of these.

My current review and comments are informed both by the present manuscript submitted to Nature Communications, as well as the response to my previous review of the original manuscript submitted to Nature Photonics, as well as consideration of the authors’ response to some of the points made by the other referee in their review of the original manuscript submitted to Nature Photonics.

Based on the entirety of my review I would recommend that the present manuscript needs to be revised based on the comments below.

Overall scientific comments:

1. One issue which I thought about more and more during my reading of the revised manuscript, and which links back to both some parts of my comments on the original manuscript, as well as those of the other referee on the original manuscript, is in terms of the identification of the origin of the visible PL emission which is the subject of the present work. Specifically, the authors are adamant that the emission is intrinsic emission, originating from the self-trapping of a free exciton, rather than emission associated with a defect. The first question then must be, why is this strong visible emission not seen in all ZnO samples? There are plenty of reports of ZnO room temperature PL with dominant near UV FE emission, and little or no broadband visible emission. This includes emission from nanostructures of various sizes and morphologies. The reasons for such variation in an intrinsic emission process must be

discussed. Does the branching ratio depend on nanostructure size, shape, surface termination etc.? The latter point also relates back to my query below on surface effects. The authors state that the absence of evidence in their electron microscopy support the identification as an intrinsic emission, but absence of evidence is not evidence of absence. And electron microscopy is not ideally suited to identifying point defects, because (a) the images are formed by passing through a finite volume of material, so the signal is an average along the path through the material, and also (b) since only a small sample of material is generally studied, which is not necessarily representative when looks for low concentrations of point defects. The authors also say that the linearity of the dependence of emission intensity on excitation power also supports the identification as an intrinsic emission, and that defect-related emission shows a sublinear effect effectively due to saturation of defect emission. However, the extent of the sub-linearity is dependent on the degree of saturation, which depends on the local fluence and a number of other parameters. Ultimately the dependence of emission intensity on excitation power for defect emission is very often extremely close to linear, and essentially indistinguishable from linear behaviour. So I don't think the identification of the visible emission is as strong as the authors suggest. See also point 5 below under detailed comments. It is worth pointing out that many of the features claimed for the emission in this work can be shown by defect-related emissions; for example a bound exciton could possibly locally distort its lattice environment and trap itself even more deeply at a defect site with a large value of Huang-Rhys factor and broad emission band, and the singlet triplet behaviour is also possible for defect-related emissions. A self-trapping process of a bound exciton could also be strongly dependent on the overall nanocrystal size and morphology, and might also affect (reduce) the interaction with the surface. I am not saying I completely disagree with the authors' identification, but rather that I think other options need to be carefully considered in the context of the results in this work and also in the context of the wider literature relating to ZnO PL.

Responds: Your professional comment is sincerely appreciated. We must admit the reported visible emission in ZnO is not exclusively for our work, the broadband visible emission has been found in other ZnO with varied sizes and morphologies (ZnO nanocrystals, ZnO nano-cones and nano-rods, ZnO films and ZnO microwires)¹⁻⁹. In the next section, the visible emission characteristics of bulk ZnO microfilaments are

studied and discussed. All previous works attributed the visible emission to the presence of impurities and structural defects^{5, 8}. Yes, the visible emission branching ratio depends on nanostructure size and shape as shown in above references, the influence of surface termination is discussed in the detailed comment (comment 2). This is because that STE formation is believed to have a strong relationship with the dimensionality of the system. In other words, a large electron-phonon coupling is crucial to the formation of STE, and the low crystal dimensionality is favor of octahedral distortion and STE formation, as well as high emission efficiency because of the strong quantum confinement particularly in the 0 D crystal structure^{10, 11}. As discussed before, a reduced dimensionality and localized electrons and holes contribute to the formation of an efficient STE. We agree totally with you that the transmission electron microscopy is a not ideally suitable method to identify point defects, but confirming the good crystallinity of ZnO nanocrystals. Yes, defect-induced emission may show linear variations versus excitation power in a certain range, while excitonic emission exhibits linear or slightly super-linear behavior. That is because the defect-induced emission is easier to reach saturation under high excitation power. It has been widely documented that if the integrated intensity of the broad PL depended linearly on excitation intensity over 3 orders of magnitude, the emission does not arise from permanent material defects^{12, 13}. As an alternative to permanent defects, photogenerated electrons/holes can couple to lattice distortions and be stabilized as “self-trapped” carriers that act as “excited-state defects”. Although further studies are required to confirm this mechanism, the linear power dependence, and the insensitivity of emission band shape to excitation intensity are consistent with emission from such photogenerated trap states. We have considered this model carefully and verified this model in detail.

2. Related to the point above, and the one in my opening paragraphs, the authors should discuss the relationship of the emission observed in their nanocrystals to the other visible emission bands typically observed in more “bulk-like” ZnO crystals (i.e. with crystal sizes much greater than the exciton Bohr radius of ~2.3 nm).

Responds: Thank you for your professional suggestions. Here, ZnO microwires with diameter of about 2 μm were used to explore the relationship of the emission observed in these nanocrystals (590 nm) to other visible emission bands (520 nm) observed in

ZnO microwires. Digital photos of ZnO microwires under the sunlight and UV excitation were shown in Figure R1a, and bright green emission can be observed under excitation. SEM images of the ZnO microwires indicate they have a uniform thickness, with a diameter of less than 10 μm (Figure R1b-1c). EDS measurement clarified that the microwires were composed of zinc and oxygen, which also demonstrated that ZnO microwires were synthesized (Figure R1d). In addition, the TEM and HRTEM images indicate a diameter of about 2 μm and high crystallinity of ZnO microwires (Figure R1e-1f). These ZnO microwires exhibited single-phase and single-crystal wurtzite structure with a preferred (002) orientation without any other peaks (Figure R1g). Additionally, the PL excitation (PLE) spectrum shows that the optimal excitation peak is located at 375 nm, and the corresponding emission spectrum is composed of a strong band peaking at 520 nm (Figure R1h). Thus, large Stokes shift (0.93 eV) and broad spectrum (full width at half-maximum (FWHM) of 140 nm, 0.66 eV) in ZnO microwires were observed, which was like typical STE emission feature of ZnO nanocrystals. 3D contour plots spectra of emission versus excitation are shown in Figure R1i, broad emission range and excitation-independent characterizations are clearly observed. For emission from 420 to 700 nm, the excitation spectra have identical shapes and features, suggesting that the broad emission originates from the recombination of the same initial excited state (Figure R1j). Additionally, light with wavelength over 400 nm cannot induce emission of the ZnO microwires in visible region from PL excitation spectrum, indicating that below-exciton or sub-gap do not contribute to the emission from permanent defects. As mentioned before, strong electron-phonon coupling is essential for the STE formation, which is evaluated by the well-known Huang-Rhys factor (S). The electron-phonon coupling has a strong connection with the FWHM of luminescence, as described by the equation:

$$\text{FWHM} = 2.36\sqrt{S}\hbar\omega_{\text{phonon}}\sqrt{\coth(\hbar\omega_{\text{phonon}}/2k_{\text{B}}T)} \quad (\text{R1})$$

where \hbar is reduced Planck constant. ω_{phonon} is phonon frequency. T is temperature, and k_{B} is Boltzmann constant. Thus, temperature-dependent PL of the ZnO microwires was measured to calculate Huang-Rhys factor S. As the temperature decreases from 300 K to 80 K, the FWHM of the broad emission decreases (Figure R1k and R1l). Huang-

Rhys factor S value of 17.3 is thus obtained, the result is lower than that of ZnO nanocrystals (73.2), because nanoparticles are more prone to lattice distortion than bulk. The phonon energy of the excited states was 48.29 meV, being close to the value of ZnO nanocrystals. The calculated STE binding energy of ZnO microwires is ~ 86.23 meV by fitting the curve of the relationship between temperature and integrated PL intensity (Figure R1m). Furthermore, the broad emission from ZnO microwires versus excitation powder over 3 orders of magnitude shows linear dependence on the excitation intensity (Figure R1n and R1o). All the evidence points to STE emission of ZnO microwires, although they show different emission peaks (520 nm) to ZnO nanocrystals (590 nm). Thus, we can infer that the STE mechanism is not exclusively for ZnO nanocrystals, it can be used in interpreting the common emission observed in bulk ZnO. Even so, we do not claim that all the other emission bands in ZnO nanoparticles or bulk ZnO can be explained by this model, it still should be investigated and verified seriously when this model were introduced.

Figure R1. (a) Digital photos of ZnO microwires under the sunlight and UV light excitation. (b) and (c) SEM images of the ZnO microwires. (d) The EDS images of O and Zn element in ZnO microwires. (e) TEM images of ZnO microwires. (f) High-resolution TEM images of ZnO microwires. (g) XRD pattern of ZnO microwires. (h) The PLE and PL spectra of ZnO microwires. (i) 3D contour plots spectra of emission versus excitation for ZnO microwires. (j) The PLE spectra of different emission

wavelengths. (k) Temperature-dependent PL spectra of ZnO microwires (80 K-300 K). (l) The FWHM of emission spectrum as a function of temperature, and the solid line is the fitting result. (m) Temperature-dependent integrated STE emission intensity of ZnO microwires. (n) PL spectra of the ZnO microwires under different power 355nm laser excitation at 300 K (excitation power: 0.2-254.3 μ W). (o) Emission intensity versus excitation power for the ZnO microwires.

Overall presentation comments:

1. While the manuscript is generally quite well written and improved from the original version, some further editing and polishing would help.

Responds: We are incredibly grateful for your careful reading and insightful suggestions and very constructive comments. The manuscript has been further edited and polished to improve its clarity.

Detailed comments:

1. A point I had mentioned in my first review was the use of the term “water solubility” or the like, when what is meant is “dispersability in water”. This issue still appears at lines 104, 189 and 467.

Responds: Thanks for your good suggestions, we have changed “water solubility” to “dispersability in water” in the line 104 and 486 of the revised manuscript (MS), and the MS has been revised thoroughly.

2. The very high PL QY of the synthesized nanocrystals in this study is one of the most notable features of the work. The lack of a degrading influence of the surfaces of the nanocrystals is particularly notable in this regard. I would like the authors to discuss the physical mechanisms which might contribute to such an unusual situation. I note that in the response to the referees’ comments on the original manuscript that the authors mentioned that ball milling had no noticeable effect on the PL properties, but the reasons why this might be the case need to be discussed. The authors also mention a silica coating can passivate surface defects. I asked about this latter point in my first review and the authors again said that ball milling had no effect and that the effect of surface states on emission had been eliminated. But it is completely unclear to me where a silica coating was applied to the present nanocrystals. When I look at the nanocrystal

synthesis information there is no mention of silica encapsulation. And also, the XPS data in figure S2(f) doesn't seem to show Si emission. This point needs to be discussed in better detail. For example, do the authors have any idea as to the nature of the ZnO nanocrystal surfaces (e.g. oxygen- or zinc-rich)?

Responds: In our manuscript, the main effect of the silica coating is to endow the ZnO nanocrystals with better dispersibility in water in terms of biological imaging. The formation of silica is due to the hydrolyzation of (3-aminopropyl)triethoxysilane (APTES) under alkaline condition, the detailed synthesis process is shown in Supporting Information. As shown in Figure R2a, the new peaks at 2936 and 1024 cm^{-1} appear in the FTIR spectrum of ZnO nanocrystals and are attributed to the characteristic stretching vibration of C-H and Si-O. In addition, N (surface $-\text{NH}_2$) and Si signal peaks observed in XPS spectra indicate the existence of surface silica (Figure R2b and Figure R2c). The above results confirm that ZnO nanocrystals with the silica coating have been successfully prepared. The ball milling does not lead to the size reduction of ZnO nanocrystals, but destroy the surface silica, resulting in decreased dispersibility in water (Figure R3). To analyze the effects of surface states, the average absolute PLQYs of ZnO nanocrystals before and after coating silica tested three times were 55.2% and 60.5% (Figure R4). The PLQYs of ZnO nanocrystals do not increase or decrease significantly after coating the silicon shell, indicating that the surface states do not play a dominant role in the broad PL emission of ZnO nanocrystals. The average ratio of Zn to O atoms in three ZnO nanocrystals samples by XPS measurement was 0.99, which was close to 1 (Table R1). In addition, electron paramagnetic resonance (EPR) spectroscopy is one of an effective method for determining the presence of impurities and native defects in solid state materials⁸. The EPR signal with g-factor close to the free-electron value (2.0023) is generally due to an unpaired electron trapped on an oxygen vacancy site (1.9965, 1.9948, 2.0190 or 2.0106)⁸. In our case, no obvious EPR signal in the self-prepared ZnO nanocrystals is observed around g-factor 2.0000 (Figure R5). This indicates that the surface of ZnO nanocrystals should be oxygen-rich.

Figure R2. (a) FTIR spectra of ZnO nanocrystals with and without silica coating. (b) XPS spectra of ZnO nanocrystals with and without silica coating. (c) Si 2p XPS spectra in ZnO nanocrystals with and without silica coating.

Figure R3. (a) TEM images of ZnO nanocrystals (inset: high-resolution TEM image). (b) TEM images of ZnO nanocrystals after ball milling (inset: high-resolution TEM image). (c) Digital photos of ZnO nanocrystals in the solid (left) and ZnO powers (right) obtained by ball milling ZnO nanocrystals under the sunlight and UV light excitation. Digital photos of ZnO nanocrystals aqueous solution before and after ball milling under the sunlight and UV light excitation.

Figure R4. The PL QY of ZnO nanocrystals tested for five times before and after coating the silicon shell.

Table R1 The ratio of Zn to O atoms in three ZnO nanocrystals samples by the XPS measurement.

Sample	Zn/O ratio	Average ratio of Zn/O
ZnO nanocrystals 1	0.92	
ZnO nanocrystals 2	1.00	0.99
ZnO nanocrystals 3	1.04	

Figure R5. The EPR spectra of ZnO nanocrystals tested for five times at room temperature.

3. In my first review of the original manuscript I mentioned the nature of the ZnO valence bandedge states and that they form from spin-orbit coupled atomic states. I also asked that this be dealt with in more detail in the revised manuscript. This is important because the spin orbit coupling process is indicated as the original of triplet to singlet conversion. The authors do provide some discussion of this in the response document to the original comments, but ultimately they revert to treating the hold as a simple spin $\frac{1}{2}$ particular. However it is more correctly described as a doublet split off from a $J = 3/2$ spin orbit coupled manifold and this means that some of the effects of the spin orbit coupling interactions are built into the correct wavefunctions. I think the authors need to revisit this section of the manuscript.

Responds: Thanks for your kind advice. Spin orbit coupling interactions are considered for construction correct wavefunctions. Since we know the wavefunction should separate, the wavefunctions can be described as: $\Phi = \varphi_{space} \times \varphi_{spin}$,

where Φ is total wavefunctions, φ_{space} is space wavefunction, φ_{spin} is spin wavefunction. For construction of wavefunction Φ , we should construct φ_{space} and φ_{spin} firstly. As we know that the valence band of ZnO are filled with electrons (ground state), O 2p and Zn 4s states are taken as ground state and excited state for intuitive understanding. Thus, we can use ‘two particles’ model to deal with this issue. For the ground state, the total orbit angular L is 2, 1, 0, and the total spin angular S is 0. For L-S coupling, the total angular J is 2,1,0, (J=3/2 is more suitable for ‘single particle’ model) and the detailed information is listed in Table R2. For the excited state, the total orbit angular L is 1, and the total spin angular S is 1, 0, and the detailed information is listed in Table R3. The overall wavefunction is always antisymmetric by construction, we can make any linear combinations of space and spin wavefunctions, and we still obtain an eigenstate. In addition, we should make sure that all the space and spin wavefunctions are individually normalized in order to construct the wavefunctions, as shown in Figure R6. Because the Hamiltonian only depends on spatial variables and not spin, we can conclude the triplet states are degenerate states.

Here, we discuss the triplet state wavefunctions caused by spin-orbit coupling and construct the overall wavefunctions based on space and spin wavefunctions. This content has been added in the revised manuscript.

Table R2. The angular of ground state and the corresponding configurations.

	S=0	Configuration
L=2	J=2	1D_2
L=1	J=1	1P_1
L=0	J=0	1S_0

Table R3. The angular of excited state and the corresponding configurations.

	L=1	Configuration
S=1	J=2,1,0	$^3P_{210}$
S=0	J=1	1P_1

Ground state	$\frac{\varphi_{space}(r_1, r_2)}{\phi_{2p}(r_1)\phi_{2p}(r_1)} \times \frac{\varphi_{spin}(\sigma_1, \sigma_2)}{\frac{1}{\sqrt{2}}(\alpha(\sigma_1)\beta(\sigma_2) - \alpha(\sigma_2)\beta(\sigma_1))}$
Singlet state	$\frac{1}{\sqrt{2}}(\phi_{2p}(r_1)\phi_{4s}(r_2) + \phi_{2p}(r_2)\phi_{4s}(r_1)) \times \frac{1}{\sqrt{2}}(\alpha(\sigma_1)\beta(\sigma_2) - \alpha(\sigma_2)\beta(\sigma_1))$
Triplet state	$\frac{1}{\sqrt{2}}(\phi_{2p}(r_1)\phi_{4s}(r_2) - \phi_{2p}(r_2)\phi_{4s}(r_1)) \times \frac{1}{\sqrt{2}}(\alpha(\sigma_1)\beta(\sigma_2) + \alpha(\sigma_2)\beta(\sigma_1))$
	$\frac{1}{\sqrt{2}}(\phi_{2p}(r_1)\phi_{4s}(r_2) - \phi_{2p}(r_2)\phi_{4s}(r_1)) \times \alpha(\sigma_1)\alpha(\sigma_2)$
	$\frac{1}{\sqrt{2}}(\phi_{2p}(r_1)\phi_{4s}(r_2) - \phi_{2p}(r_2)\phi_{4s}(r_1)) \times \beta(\sigma_1)\beta(\sigma_2)$

Figure R6. The wavefunctions of the excited state. ϕ_{2p} : Space wavefunction in 2p orbit; ϕ_{4s} : Space wavefunction in 4s orbit; α : Spin up wavefunction; β : Spin down wavefunction; σ_1 : Particle 1; σ_2 : Particle 2.

4. The authors do address my comment in my review of the original manuscript concerning the notable peak in the 4 photon absorption cross-section at ~ 1150 nm, which they say is due to a higher single photon absorption cross-section at 287.5 nm, combined with the relatively high absorption of ethanol solvent. They also show a figure, but to my mind, the data in the figure for the single photon absorption cross-section is quite continuous and doesn't show evidence of any peak (acknowledging that the y-axis is a logarithmic scale). Is it possible to utilize a different solvent which doesn't have an overlapping absorption onset?

Responds: The notable peak in the four-photon absorption cross-section at ~ 1150 nm is due to the larger step size (50 nm) for the test of multi-photon absorption cross section, this has been widely observed when researchers recorded three-photon, four-photon

and five-photon absorption cross section of other luminescence materials, such as MAPbBr₃, CsPbBr₃, and MAPbBr₃/(OA)₂PbBr₄ perovskite nanocrystals^{14, 15} (Figure R7). The peak in the four-photon absorption cross-section at ~ 1150 nm is in the acceptable scope after correction. In addition, we have selected several common solvents (such as ethanol, water, toluene, dichloromethane, and n-hexane) for absorption spectrum characterization in the range of 800 nm-1600 nm. As shown in Figure R8, these solvents all exhibit varying degrees of absorption in 800 nm-1600 nm range. In this experiment, ethanol was selected as the solvent for testing the absorption cross section. In addition, the change of laser power after passing ethanol have been measured to exclude the effect of solvent. Firstly, the laser power before and after passing through the solvent in the wavelength range of 1150-1600 nm were measured to calculate the transmittance of the solvent (Figure R9). According to the transmittance of the solvent at different excitation wavelength, the laser power acting on the sample was obtained and shown in Table R4. The multi-photon absorption cross-section (σ_n) was recalculated as shown in Figure R10. The content has been added in the revised manuscript.

Figure R7. Three-photon, four-photon and five-photon action cross-section spectra of the MAPbBr₃, CsPbBr₃ and MAPbBr₃/(OA)₂PbBr₄ perovskite nanocrystals¹⁴.

Figure R8. The absorption spectra of ethanol, water, toluene, dichloromethane, and n-hexane.

Figure R9. The transmittance of the ethanol solvent at different excitation wavelengths (1150 nm, 1200 nm, 1250 nm, 1300 nm, 1350 nm, 1400 nm, 1450 nm, 1500 nm, and 1600 nm).

Table R4. Parameters required for the calculation of multiphoton absorption cross section (λ : The excitation wavelength, P: The laser power, a: Incident spot diameter, F: The integrated intensity of emission spectra. I_0 : Photon density per unit time.

λ (nm)	P (mW)	a (cm)	F	I_0 (photon s ⁻¹ cm ²)
800	11	0.037	130568.197	2.05E+28
850	12.7	0.040	185109.0037	2.50E+28

900	12.9	0.040	135923.2387	2.44E+28
950	10.5	0.039	54586.19772	1.78E+28
1000	9.7	0.053	112345.2467	2.93E+28
1050	12.4	0.062	47060.44476	4.87E+28
1100	22.5	0.039	205480.9488	3.36E+28
1150	14.0	0.039	159913.5489	1.98E+28
1200	29.5	0.036	198461.9151	3.46E+28
1250	35.3	0.036	991524.3264	3.99E+28
1300	35.4	0.038	935478.0272	4.17E+28
1350	27.3	0.040	198296.7557	3.44E+28
1400	37.0	0.042	573632.3892	4.95E+28
1450	27.6	0.042	272296.3	3.52E+28
1500	19.4	0.042	85678.94369	2.45E+28
1550	21.5	0.042	177528.6233	2.63E+28
1600	31.1	0.046	115068.0572	4.38E+28

Figure R10. Three-photon absorption cross-sections (σ_3), four-photon absorption cross-sections (σ_4) and five-photon absorption cross-sections (σ_5) of the ZnO nanocrystals in the wavelength range 800-1600 nm at room temperature.

5. The PLE data in figure 2(b) are interesting, in that wavelengths shorter than ~ 360 nm are very inefficient in exciting the visible emission band. This is quite different to the emission of the ZnO microwires in figure S10(h). If the emission is an intrinsic band,

and not defect-related, then why should the PLE vary in this way?

Responds: Thank you for your professional comments. PL excitation spectra show the change in fluorescence intensity as a function of the wavelength of the excitation light and are measured using a spectrofluorometer. The PLE signal can be described as the ratio of the emitted photon flux I_{emit} and the incident excitation photon flux I_{excite} . The emission intensity of ZnO nanocrystals increases firstly then decreases little until toward an unchanging equilibrium, as shown in Figure R11. Thus, the emitted photon flux I_{emit} under the excitation of ~ 360 nm steps to that of under excitation wavelength shorter than 360 nm. Namely, the ratios of $I_{\text{emit}} / I_{\text{excite}}$ at different excitation wavelengths become less noticeable, leading to the difference in PLE spectra in different samples. This can be convinced by the excitation-emission contour plots of ZnO nanocrystals aqueous solution, powders and ZnO microwires (Figure R12).

Figure R11. (a) The PL spectra of ZnO nanocrystals with different concentrations (0.2 mM-192 mM). (b) The PL intensity values of ZnO nanocrystals with different concentrations.

Figure R12. (a) Emission contour plots of ZnO nanocrystals solution as excitation wavelength varies from 300-600 nm at room temperature. (b) Emission contour plots of ZnO nanocrystals powder. (c) Emission contour plots of ZnO microwires.

of the ZnO nanocrystals powders as excitation wavelength varies from 300-600 nm at room temperature. (c) Emission contour plots of the ZnO microwires as excitation wavelength varies from 300-600 nm at room temperature.

Reviewer #2 (Remarks to the Author):

In the revised manuscript, the authors have well addressed most of my questions. The manuscript may be suitable for publication in nature communications. However, I still have several questions to be answered by the authors.

1. The authors claimed that singlet/triplet mixed exciton emission was responsible for the high emission efficiency of visible emission and the split energy between singlet and triplet exciton of 58 meV. It is well known that small split energy will be favorable for the occurrence of RISC. In addition, 58 meV is even much larger than the electron energy at 300 K (approximately 26 meV). In their case, How do electrons overcome this energy difference? The authors must well address this question and give the discussions in the text.

Responds: Your comment is appreciated. The split energy between singlet and triplet exciton of ZnO nanocrystals was about 60 meV obtained from the temperature-dependent average lifetime, temperature-dependent PL spectra and the theoretical calculations. Generally, the split energy was the difference between the lowest energies of the singlet and triplet states. But, as shown in the Jablonski diagram, potential energy is different at different nuclear coordinate (Figure R13a), thus, the RISC process from triplet to singlet state could happen with different rates at different temperatures. According to the result of the temperature-dependent PL-decay dynamics of ZnO nanocrystals from 80-300 K, the emission long lifetime (τ_2) microsecond scale suddenly appeared at the temperature up to 160 K (Figure R13b and Figure R13c). This result is mainly attributed to different potential energies at different coordinates. In this work, an obvious RISC onset process was occurred at about 160 K, corresponding to 13.9 meV, as shown in Figure R13d. The content has been discussed and added in the revised manuscript.

Figure R13. (a) The Jablonski diagram¹⁶. (b) Temperature-dependent PL lifetime decay spectra of ZnO nanocrystals. (c) Temperature-dependent PL lifetime of the ZnO nanocrystals (τ_1 : the short lifetime; τ_2 : the long lifetime). (d) The schematic diagram of singlet/triplet mixed STE emission mechanism of ZnO nanocrystals.

2. The authors claimed “it is not surprising that 3PA cross-sections of ZnO nanocrystals are much smaller than that of MAPbBr₃, while 4PA and 5PA cross-sections are comparable with MAPbBr₃.” However, I’m still not satisfied with this response. Actually, the single intermediate state (SIS) approximation can be used to give an order-of-magnitude estimation (C. Xu and W. W. Webb, Topics in Fluorescence Spectroscopy, Vol. 5 (Springer, 1997), Chap. 11). The results of the authors deviate significantly from the predicted values. I strongly suggested the authors use other technique to determine the multiphoton absorption cross sections, such as Z-scan technique.

Responds: Thank you for your comment and recommendation. Open-aperture Z-scan technique has been applied for measuring the multiphoton absorption cross-sections of ZnO nanocrystals, according to your suggestion. The open-aperture Z-scan requires very high excitation peak intensity due to its low detection efficiency in the characterization of higher-order multiphoton absorption (such as four-photon and five-photon absorption)¹⁴. But the extremely high excitation intensities may result in sample damage or introduce other effects (or artifacts). Therefore, the multiphoton absorption

(such as four-photon and five-photon absorption) were performed by combining multi-photon excited PL measurements with open-aperture Z-scan technology. In our manuscript open-aperture Z-scan has been performed at 800 nm in view of negligible absorption in this region (Figure R8), and multiphoton absorption cross-sections of ZnO nanocrystals at other wavelengths were obtained by utilizing the three-photon absorption cross sections at 800 nm measured by Z-scan as the standard. The three-photon absorption cross sections of ZnO nanocrystals at 800 nm were tested by the open-aperture Z-Scan method under 289.6 μW , 305.1 μW and 329.9 μW excitation power. The three-photon absorption fitting formulas were as follows¹⁴:

$$z_0 = \pi\omega_0^2/\lambda \quad (\text{R2})$$

$$L'_{eff} = [1 - \exp(-2\alpha_0 L)]/2\alpha_0 \quad (\text{R3})$$

$$I_0 = I_{00}/(1 + z^2/z_0^2)^2 \quad (\text{R4})$$

$$T(z) = 1 - \frac{\alpha_3 I_0^2 L'_{eff}}{3^{3/2} (1 + z^2/z_0^2)^2} \quad (\text{R5})$$

Where L'_{eff} is the effective sample lengths for three-photon absorption processes; L is the sample length; I_0 is the excitation intensity at position z , I_{00} is the excitation intensity at position z_0 ; α_3 is the three-photon absorption coefficient; z_0 is the Rayleigh length; ω_0 is the minimum beam waist at the focal point ($z = 0$); λ is the laser free-space wavelength; α_0 is the linear absorption coefficient, α_3 is the three-photon absorption coefficient; $T(z)$ is the normalized energy transmittance at position z . As shown in Figure R14, the three-photon absorption cross sections (σ_s) at 800 nm were calculated as $0.5 \times 10^{-78} \text{ cm}^6 \text{ s}^2 \text{ photon}^{-2}$, $0.93 \times 10^{-78} \text{ cm}^6 \text{ s}^2 \text{ photon}^{-2}$ and $0.69 \times 10^{-78} \text{ cm}^6 \text{ s}^2 \text{ photon}^{-2}$ by the formula R5, respectively. The average three-photon absorption cross section at 800 nm was $0.71 \times 10^{-78} \text{ cm}^6 \text{ s}^2 \text{ photon}^{-2}$, which is in the same order of magnitude with that of ZnO nanocrystals measured through multiphoton-excited PL technique. The three-photon cross section of the nanocrystals excited by other wavelengths (900, 1000, 1050 and 1100 nm) are obtained by the following formulas¹⁷:

$$F_3/F_s = I_3^3 \sigma_3 / I_s^3 \sigma_s \quad (\text{R6})$$

Where F_3 , I_3 and σ_3 were the integrated intensity of the three-photon emission spectrum, the photon density per unit time and the three-photon cross section of ZnO nanocrystals

at different excitation wavelength (850, 900, 950, 1000, 1050 and 1100 nm), respectively. F_s , I_s and σ_s was the integrated intensity of the three-photon emission spectrum, the photon density per unit time and the three-photon cross section of ZnO nanocrystals at 800 nm. In addition, the four-photon (σ_4) and five-photon cross section (σ_5) of the sample were determined by these following equations¹⁴:

$$F_4/F_s = I_4^4 \sigma_4 / I_s^3 \sigma_s \quad (R7)$$

$$F_5/F_s = I_5^5 \sigma_5 / I_s^3 \sigma_s \quad (R8)$$

where F_4 , I_4 and σ_4 were the integrated intensity of the four-photon emission spectrum, the photon density per unit time and the four-photon cross section of ZnO nanocrystals under different excitation wavelength (1150, 1200, 1300, 1400 and 1450 nm), respectively. F_5 , I_5 and σ_5 were the integrated intensity of the five-photon emission spectrum, the photon density per unit time and the five-photon cross section of ZnO nanocrystals at different excitation wavelength (1500, 1550 and 1600 nm), respectively. And through applying the above equations R6-R8 and utilizing the three-photon absorption cross section σ_s at 800 nm measured by Z-scan as the standard, the σ_n ($n=3, 4, \text{ and } 5$) values of ZnO nanocrystals can be obtained over the wavelength range spanning from 850-1600 nm (Figure R10). The content has been added in the revised manuscript.

Figure R14. Open-aperture Z-Scan experimental curves and theoretically fitted curves of ZnO nanocrystals under 800 nm fs laser excitation.

3. For the sudden maximum at 1150 nm in Figure 5c, the authors claimed it was caused

by the higher single photon absorption at 287.5 nm and the relatively high absorption of ethanol solvent. I'm still not convinced by this explanation. It is obvious from the data given by the authors that the profiles of multiphoton and linear absorption spectra are quite different. In addition, the authors clearly knew that there is a significant absorption of solvent, why not try to eliminate this effect?

Responds: Thank you for your kind comment. The effect of absorption of solvent has been eliminated and corrected in the revised manuscript and supporting information. Firstly, the laser power before and after passing through the solvent in the wavelength range of 1150-1600 nm were measured to calculate the transmittance of the solvent (Figure R8 and R9). According to the transmittance of the solvent under the different excitation wavelength, the laser power acting on the sample was obtained and shown in Table R4. In addition, the three-photon absorption cross section at 800 nm was performed with open-aperture Z-scan. The multi-photon absorption cross-sections (σ_n) of ZnO nanocrystals were obtained by combining Z-scan and multiphoton excited PL technique, as shown in Figure R10. The peak in the four-photon absorption cross-section at ~ 1150 nm is in the acceptable scope after correction, in view of the larger step size (50 nm) for the test of multi-photon absorption cross section (Figure R7).

Reference

1. Cheng, W. D., Wu, P., Zou, X. Q. & Xiao, T. Study on synthesis and blue emission mechanism of ZnO tetrapodlike nanostructures. *J. Appl. Phys.* **100**, 054311-054311-4 (2006).
2. Fu, Y. S. et al. Stable Aqueous Dispersion of ZnO Quantum Dots with Strong Blue Emission via Simple Solution Route. *J. Am. Chem. Soc.* **129**, 16029-16033 (2007).
3. Shi, H. Q. et al. Synthesis of silane surface modified ZnO quantum dots with ultrastable, strong and tunable luminescence. *Chem. Comm.* **47**, 11921-11923 (2011).
4. Singh, K., Chaudhary, G.R., Singh, S. & Mehta, S.K. Synthesis of highly luminescent water stable ZnO quantum dots as photoluminescent sensor for picric acid. *J. Lumin.* **154**, 148-154 (2014).

5. Kumar, V. et al. Origin of the red emission in zinc oxide nanophosphors. *Mater. Lett.* **101**, 57-60 (2013).
6. Zhao, D. et al. Luminescent ZnO quantum dots for sensitive and selective detection of dopamine. *Talanta* **107**, 133-139 (2013).
7. Chen, Z. et al. Effect of annealing on photoluminescence of blue-emitting ZnO nanoparticles by sol–gel method. *J. Solgel Sci. Technol.* **62**, 252-258 (2012).
8. Zeng, H. et al. Blue Luminescence of ZnO Nanoparticles Based on Non-Equilibrium Processes: Defect Origins and Emission Controls. *Adv. Funct. Mater.* **20**, 561-572 (2010).
9. Xiong, H. M. ZnO Nanoparticles Applied to Bioimaging and Drug Delivery. *Adv. Mater.* **25**, 5329-5335 (2013).
10. Li, S., Luo, J., Liu, J. & Tang, J. Self-Trapped Excitons in All-Inorganic Halide Perovskites: Fundamentals, Status, and Potential Applications. *J. Phys. Chem. Lett.* **10**, 1999-2007 (2019).
11. Jun, T. et al. Lead-Free Highly Efficient Blue-Emitting Cs₃Cu₂I₅ with 0D Electronic Structure. *Adv. Mater.* **30**, e1804547 (2018).
12. Luo, J. J. et al. Efficient and stable emission of warm-white light from lead-free halide double perovskites. *Nature* **563**, 541-545 (2018).
13. Dohner, E. R., Jaffe, A., Bradshaw, L. R. & Karunadasa, H. I. Intrinsic white-light emission from layered hybrid perovskites. *J. Am. Chem. Soc.* **136**, 13154-13157 (2014).
14. Chen, W. et al. Giant five-photon absorption from multidimensional core-shell halide perovskite colloidal nanocrystals. *Nat. Commun.* **8**, 15198 (2017).
15. Yu, J. H. et al. High-resolution three-photon biomedical imaging using doped ZnS nanocrystals. *Nat. Mater.* **12**, 359-366 (2013).
16. Westermayr, J. & Marquetand, P. Machine Learning for Electronically Excited States of Molecules. *Chem. Rev.* **121**, 9873-9926 (2021).

17. Jin, C. et al. Rational Design of Cyclometalated Iridium (III) Complexes for Three-Photon Phosphorescence Bioimaging. *Angew. Chem. Int. Ed.* **59**, 15987-11599 (2020).

REVIEWERS' COMMENTS

Reviewer #2 (Remarks to the Author):

The authors have well answered my questions. The current manuscript can be accepted.

Reviewer #3 (Remarks to the Author):

The changes that the authors have made in revising the manuscript have improved the work and addressed the majority of the issues I raised in my previous reviews.

I would suggest that the authors add some comment to the manuscript to clarify the reason for the dominant visible STE emission in the present samples and the reasons for variation in the near UV FE emission to visible emission ratio in different samples. The authors mention plausible reasons in their rebuttal, including that the visible emission branching ratio depends

on nanostructure size and shape and that STE formation is believed to have a strong relationship with the dimensionality of the system. Also, a large electron-phonon coupling is crucial to the formation of STE, and the low crystal dimensionality favours local distortion and STE formation, as well as high emission efficiency because of the strong quantum confinement.

Overall, I believe that the manuscript is now suitable for publication in Nature Communications, but would suggest a small expansion of the manuscript to include the point above.

Response to Reviewers' comments

Reviewer #2 (Remarks to the Author):

The authors have well answered my questions. The current manuscript can be accepted.

Responds: We sincerely thank you for agreeing to publish the current manuscript in *Nature Communications*.

Reviewer #3 (Remarks to the Author):

The changes that the authors have made in revising the manuscript have improved the work and addressed the majority of the issues I raised in my previous reviews.

I would suggest that the authors add some comment to the manuscript to clarify the reason for the dominant visible STE emission in the present samples and the reasons for variation in the near UV FE emission to visible emission ratio in different samples. The authors mention plausible reasons in their rebuttal, including that the visible emission branching ratio depends on nanostructure size and shape and that STE formation is believed to have a strong relationship with the dimensionality of the system. Also, a large electron-phonon coupling is crucial to the formation of STE, and the low crystal dimensionality favours local distortion and STE formation, as well as high emission efficiency because of the strong quantum confinement.

Overall, I believe that the manuscript is now suitable for publication in Nature Communications, but would suggest a small expansion of the manuscript to include the point above.

Responds: Thank you for your recommendation. Some comments have been added in

the lines 346-349 of the revised manuscript to clarify the reason for the dominant visible STE emission in the present samples and the reasons for variation in the near UV FE emission to visible emission ratio in different samples. Because low crystal dimensionality is more likely to occur the local lattice distortion to form STE emission. Therefore, the visible STE emission depends on nanostructure size and shape.